# Augmentation of WRF-Hydro to Simulate Overland Flow- and Streamflow-Generated Debris Flow Susceptibility in Burn Scars

**Chuxuan Li[1], Alexander L. Handwerger[2,3], Jiali Wang[4], Wei Yu[5,6], Xiang Li[7], Noah J. Finnegan[8], Yingying Xie[9,10], Giuseppe Buscarnera[7], and Daniel E. Horton[1]**

[1] Department of Earth and Planetary Sciences, Northwestern University, Evanston, IL, 60208, USA

[2] Joint Institute for Regional Earth System Science and Engineering, University of California, Los Angeles, CA, 90095, USA

[3] Jet Propulsion Laboratory, California Institute of Technology, Pasadena, CA, 91109, USA

[4] Environmental Science Division, Argonne National Laboratory, Lemont, IL, 60439, USA

[5] Cooperative Institute for Research in Environmental Sciences, University of Colorado Boulder, CO, 80309, USA

[6] NOAA/Global Systems Laboratory, 325 Broadway Boulder, Denver, CO, 80305-3328, USA

[7] Department of Civil and Environmental Engineering, Northwestern University, Evanston, IL, 60208, USA

[8] Department of Earth and Planetary Sciences, University of California Santa Cruz, Santa Cruz, CA, 95064, USA

[9] Program in Environmental Sciences, Northwestern University, 2145 Sheridan Road, Evanston, IL, 60208, USA

[10] Department of Biological Sciences, Purdue University, 915 W State St, West Lafayette, IN 47907, USA

*Correspondence to:* Chuxuan Li (chuxuanli2020@u.northwestern.edu)

**Abstract**

In steep wildfire-burned terrains, intense rainfall can produce large runoff that can trigger highly destructive debris flows. However, the ability to accurately characterize and forecast debris-flow susceptibility in burned terrains using physics-based tools remains limited. Here, we augment the Weather Research and Forecasting Hydrological modeling system (WRF-Hydro) to simulate both overland and channelized flows and assess postfire debris flow susceptibility over a regional domain. We perform hindcast simulations using high-resolution weather radar-derived precipitation and reanalysis data to drive non-burned baseline and burn scar sensitivity experiments. Our simulations focus on January 2021 when an atmospheric river triggered numerous debris flows within a wildfire burn scar in Big Sur – one of which destroyed California's famous Highway 1. Compared to the baseline, our burn scar simulation yields dramatic increases in total and peak discharge, and shorter lags between rainfall onset and peak discharge, consistent with streamflow observations at nearby U.S. Geological Survey (USGS) streamflow gage sites. For the 404 catchments located in the simulated burn scar area, median catchment-area normalized peak discharge increases by ~450% compared to the baseline. Catchments with anomalously high catchment-area normalized peak discharge correspond well with post-event field-based and remotely-sensed debris flow observations. We suggest that our regional post-fire debris flow susceptibility analysis demonstrates WRF-Hydro as a compelling new physics-based tool whose utility could be further extended via coupling to sediment erosion and transport models and/or ensemble-based operational weather forecasts. Given the high-fidelity performance of our augmented version of WRF-Hydro, as well as its potential usage in probabilistic hazard forecasts, we argue for its continued development and application in post-fire hydrologic and natural hazard assessments.

**Short Summary**

In January 2021 a storm triggered numerous debris flows in a wildfire burn scar in California. We use a hydrologic model to assess debris flow susceptibility in pre-fire and postfire scenarios. Compared to pre-fire conditions, postfire conditions yield dramatic increases in peak water discharge, substantially increasing debris flow susceptibility. Our work highlights the hydrologic model's utility in investigating and potentially forecasting postfire debris flows at regional scales.

## 1 Introduction

Following intense rainfall, areas with wildfire burn scars are more prone to flash flooding (Neary et al., 2003; Bart & Hope 2010; Bart 2016) and runoff-generated debris flows than unburned areas (Ice et al., 2004; Shakesby & Doerr, 2006; Moody et al., 2013). After wildfire, reduced tree canopy interception, decreased soil infiltration due to soil-sealing effects (Larsen et al., 2009), and increased soil water repellency – especially in hyper-arid environments (Dekker & Ritsema, 1994;

Doerr & Thomas, 2000; MacDonald & Huffman, 2004) – increases excess surface water, and on sloped terrains leads to overland flow (Shakesby & Doerr, 2006; Stoof et al., 2012). As water moves down hillslopes and erosion adds sediment to water-dominated flows, clear water floods can transition to turbulent and potentially destructive debris flows (Meyer & Wells, 1997; Cannon et al., 2001, 2003; Santi et al., 2008). In contrast to debris flows initiated by shallow landslides, this rainfall-runoff process has been identified as the major cause for postfire debris flows in the western U.S. (Cannon, 2001; Cannon et al., 2003, 2008; Kean et al., 2011; Parise & Cannon, 2012; Nyman et al., 2015) and in other regions that are particularly susceptible to wildfires and subsequent heavy precipitation (Bisson et al., 2005; Mitsopoulos & Mironidis, 2006; Rosso et al., 2007; Parise & Cannon, 2008, 2009).

On the U.S. west coast, atmospheric rivers (ARs) are the dominant synoptic weather systems responsible for producing postfire debris flows (Barth et al., 2017; Oakley et al., 2017, 2018; Young et al., 2017). ARs are long filament-like bands of elevated water vapor within the lower troposphere that often form over ocean basins. They are responsible for over 90% of poleward water vapor transport (Zhu & Newell, 1998) and often result in heavy precipitation upon landfall, particularly with orographic uplift (Ralph et al., 2004; Neiman et al., 2008). It is reported that 30–50% of annual precipitation and 60%–100% of extreme precipitation along the U.S. west coast is the result of ARs (Collow et al., 2020; Eldardiry et al., 2019; Hecht & Cordeira, 2017). In California, anthropogenic climate change is projected to increase AR intensity (Huang et al., 2020a, 2020b), increase the intensity and frequency of wet-season precipitation (Polade et al., 2017; Swain et al., 2018), increase wildfire potential (Brown et al., 2020; Swain 2021), and extend the wildfire season (Goss et al., 2020). As such, the occurrence and intensity of postfire debris flows are likely to increase as the effects of anthropogenic climate change persist (Cannon & DeGraff, 2009; Kean & Staley, 2021; Oakley 2021).

Due to this increasing threat, the development of tools to assess postfire debris flow susceptibility and hazards is critical. However, due to long-standing terminology ambiguity in the natural hazard community (Reichenbach et al., 2018), we first begin with a definition of terms. In this study we demonstrate the use of a new physics-based tool to map postfire debris flow susceptibility at regional scales. We follow the guidance of [Reichenbach et al., (2018) & references therein] and define *susceptibility* as the likelihood of debris flow occurrence in an area, and *hazard* as the probability of debris flow occurrence of a given magnitude within a specified area and period of time. In other words, debris flow susceptibility neither simulates debris flow dynamics such as initiation nor estimates debris flow size or considers the timing or frequency of the debris flow occurrence. Rather, it focuses on locating areas prone to debris flows considering local environmental factors (Brabb 1985; Guzzetti et al., 2005).

Heuristic, deterministic, statistical approaches, and coupled deterministic and statistical models have previously been employed to assess landslide susceptibility (Dahal et al., 2008; Regmi et al., 2010; Park et al., 2016; Reichenbach et al., 2018). For postfire debris flow susceptibility or hazard assessment, however, the use of deterministic models is limited. In contrast, statistical approaches

are commonly used in both research and operational settings (Cannon et al., 2010; Friedel 2011a,
2011b; Gardner et al., 2014; Staley et al., 2016; Nikolopoulos et al., 2018; Cui et al., 2019). For
example, rainfall intensity-duration (ID) thresholds are one of the simplest-to-implement and most
widely used statistical methods for mapping rainfall-induced landslide susceptibility including
postfire debris flows (Cannon et al., 2011; Staley et al., 2017). In addition, the U.S. Geological
Survey (USGS) currently employs a statistical approach in their Emergency Assessment of
Postfire Debris-flow Hazards that consists of a logistic regression model to predict the likelihood
of post-wildfire debris flows (e.g., Cannon et al., 2010; Staley et al., 2016), and a multiple linear
regression model to predict debris flow volumes (Gartner et al., 2014). Machine-learning
techniques such as self-organizing maps, genetic programming, and a random forest algorithm
have also been used to predict postfire debris flows in the western U.S. (Friedel 2011a, 2011b;
Nikolopoulos et al., 2018). In general, statistical approaches are useful for identifying and
characterizing relationships amongst contributing environmental factors and are widely used due
to their low computational costs and the potential for rapid assessment. Despite the utility and
advantages of data-driven hazard prediction approaches over regional domains, these techniques
(1) do not simulate the underlying physics, (2) often require large amount of historical observation
data that may not be readily available, and (3) result in models that are often only applicable to
specific locales. These limitations inhibit their utility in postfire debris flow susceptibility
assessment from a physics-based perspective, limit their applicability in climatological and
geographic settings different than their training sites, and limit their use in non-stationary
conditions (e.g., under changing climatic conditions).
In contrast, physics-based models that simulate spatially-explicit hydrologic and mass wastage
processes are well-suited for sensitivity analyses in diverse settings. However, studies employing
deterministic process-based models have tended to focus on rainfall-induced shallow landslides
(Crosta & Frattini, 2003; Claessens et al., 2007) or landslide-induced debris flows (e.g., Iverson &
George, 2014; George & Iverson, 2014), rather than on runoff-generated debris flows which are
more common in postfire areas (Cannon et al., 2001, 2003; Santi et al., 2008). Studies that have
investigated postfire hydrologic responses using physics-based models have largely focused on
mechanistic studies such as short-term responses at high spatiotemporal resolutions (Rengers et
al., 2016; McGuire et al., 2016, 2017) or long-term runoff responses at coarse temporal resolutions
(McMichael & Hope, 2007; Rulli & Rosso, 2007) in individual catchments. For example, process-
based models have employed shallow water equations to better understand the triggering (McGuire
et al., 2017; Tang et al., 2019a, 2019b) and sediment transport mechanisms (McGuire et al., 2016)
of postfire debris flows as well as the timing of postfire debris flows (Rengers et al., 2016). The
numerical models employed by these studies are used to simulate debris flow dynamics rather than
assess susceptibility over regional domains, as such they focus on individual catchments (with
drainage areas of ~1 km$^2$) with very high spatiotemporal resolutions (Rengers et al., 2016;
McGuire et al., 2016, 2017; Tang et al., 2019a, 2019b). In addition to individual catchment
applications, process-based models often adopt simplifications that can limit effective prediction
and hypothesis testing to overcome computational limits. For example, the kinematic runoff and

erosion model (KINEROS2) simplifies drainage basins into 1-dimensional channels and hillslope patches (Canfield et al., 2005; Goodrich et al., 2012; Sidman et al., 2016), and the Hydrologic Modeling System (HEC-HMS) uses an empirically-based curve number method to estimate saturation excess water (Cydzik et al., 2009), which cannot resolve infiltration excess overland flow, a critical process in burn scars (Chen et al., 2013).

Given the current state of debris flow susceptibility assessment and prediction in previously burned terrains, in addition to the growing influence of anthropogenic climate change on wildfire and extreme precipitation, development of physics-based susceptibility mapping tools that can be used in both hindcast investigations and forecasting applications is needed. Furthermore, due to the diverse morphology and often large spatial scales of precipitation events and their interactions with geographically distributed wildfire burn scars, development of tools that can assess susceptibility over regional domains, particularly in operational forecasting applications, is critical. Here, to advance the field of burn scar debris flow susceptibility assessment, we explore the use of the physics-based and fully-distributed Weather Research and Forecasting Hydrological modeling system version 5.1.1 (WRF-Hydro). WRF-Hydro is an open-source community model developed by the National Center for Atmospheric Research (NCAR). It is the core of the National Oceanic and Atmospheric Administration's (NOAA) National Water Model forecasting system and has been used extensively to study channelized flows over regional domains (e.g., Wang et al., 2019; Lahmers et al., 2020). Here, we modify WRF-Hydro to output high temporal resolution fine-scale (100 m) debris flow-relevant overland flow; a process computed using a fully unsteady, explicit, finite difference diffusive wave formulation. Previous efforts, employing shallow water equations, diffusive, kinematic, and diffusive-kinematic wave models, have demonstrated that water-only models can provide critical insights on runoff-driven debris flows (Arattano & Savage, 1994; Arattano & Franzi, 2010; Di Cristo et al., 2021), even in burned watersheds (Rengers et al., 2016; McGuire & Youberg, 2020).

To test and demonstrate the utility of WRF-Hydro in debris flow studies, we investigate the January 2021 debris flow events within the Dolan burn scar on the Big Sur coast of central California (Fig. 1a–b). We first identify multiple debris flow sites using optical and radar remote sensing data and field investigations. We then calibrate WRF-Hydro against ground-based soil moisture and streamflow observations and use it to study the effects of burn scars on debris flow hydrology and susceptibility. The paper is organized as follows. Section 2 describes the identification approach and geologic setting of debris flows. Section 3 presents a description of WRF-Hydro. Section 4 describes the simulation, calibration, and validation of WRF-Hydro. Section 5 presents the results. Section 6 discusses the results and Sect. 7 provides a conclusion.

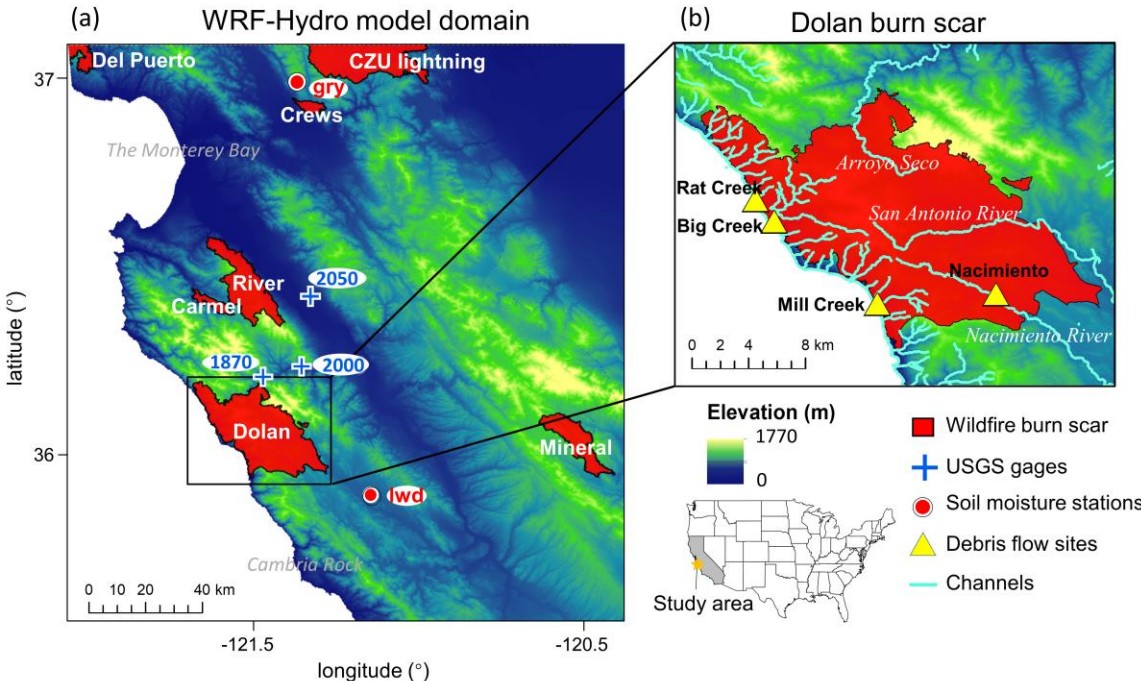

178

**Fig. 1**| WRF-Hydro model domain and Dolan burn scar. (a) WRF-Hydro model domain depicting topography, 2020 wildfire season burn scars, and PSL soil moisture and USGS stream gage observing sites. The black rectangle outlines (b) the Dolan burn scar inset, in which debris flow locations and major streams are marked and labeled. The location of the study area is shown in the embedded U.S. map with the state of California shaded in grey.

184

## 2 Study domain and debris flow identification methodology

The Dolan wildfire burned from August 18$^{th}$ till December 31$^{st}$, 2020. 55% of areas within the fire perimeter were burned at moderate-to-high severity (Burned Area Emergency Response, 2020). After the fire, USGS Emergency Assessment of Postfire Debris-flow Hazards produced a debris flow hazard assessment using a design storm based statistical model (USGS, 2020). On January 27–29, 2021, an AR made landfall on the Big Sur coast, bringing more than 300 mm of rainfall to California's Coast Ranges (Fig. 2), with a peak rainfall rate of 24 mm h$^{-1}$ [calculated with Multi-Radar/Multi-Sensor System (MRMS) precipitation; Zhang et al., 2011, 2014, 2016]. During the AR event, a section of California State Highway 1 (CA1) at Rat Creek was destroyed by a debris flow. CA1 was subsequently closed for three months and rebuilt at a cost of ~$11.5M (Los Angeles Times, 2021).

196

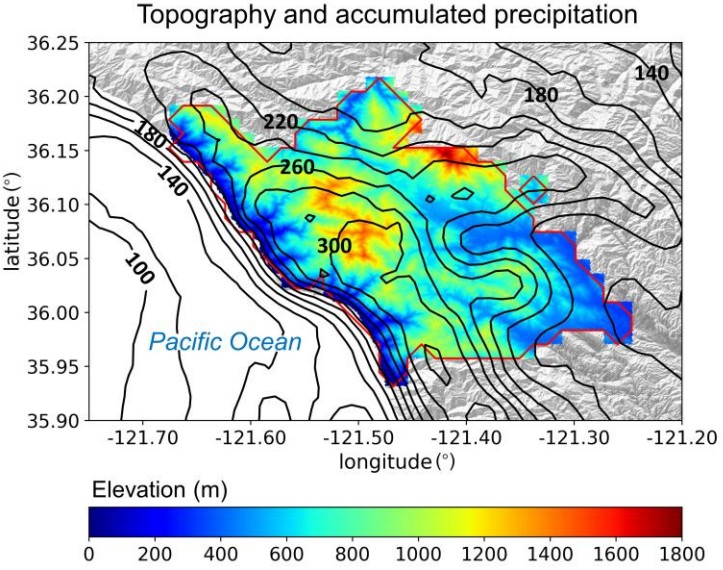

Fig. 2| The topography (m; shading) and MRMS accumulated precipitation (mm; contour lines) during the AR event from January 27$^{th}$ 00:00 to 29$^{th}$ 23:00 in the Dolan burn scar. Contour line interval for accumulated precipitation is 20 mm, and lines of 100, 140, 180, 220, 260, and 300 mm are labeled. The red polygon outlines the perimeter of the Dolan burn scar.

## 2.1 Debris flow identification from remote sensing and field work

In addition to the Rat Creek debris flow, which made national news (Los Angeles Times, 2021), we identified three other debris flows using a combination of field investigation, and open access satellite optical and synthetic aperture radar (SAR) images (Fig. 3 and Fig. B1).

We examined relative differences in normalized difference vegetation index (rdNDVI) defined by (Scheip & Wegmann, 2021):

$$rdNDVI = \frac{NDVI_{post} - NDVI_{pre}}{\sqrt{NDVI_{pre} + NDVI_{post}}} \times 100 \qquad (1)$$

where $NDVI_{pre}$ and $NDVI_{post}$ are the pre- and post-event normalized difference vegetation index (NDVI) images computed following:

$$NDVI = \frac{NIR - Red}{NIR + Red} \qquad (2)$$

where *NIR* is the near-infrared response and *Red* is the visible red response. rdNDVI was calculated from 10-m Sentinel-2 satellite data using the HazMapper v1.0 Google Earth Engine application (Scheip & Wegmann, 2021). HazMapper requires selection of an event date, pre-event window (months), post-event window (months), max cloud cover (%) and slope threshold (°). These input

requirements filter the number of images used to calculate the rdNDVI. We set the event date to
January 27[th], 2021 and used a 3 month pre- and post-event window with 0% max cloud cover and
a 0° slope threshold to identify vegetation loss associated with the debris flows. We then created
a binary map to highlight debris flows (and other vegetation loss) pixels above a rdNDVI
vegetation loss threshold. We removed all pixels with rdNDVI > -10.
Lastly, we searched for debris flows (and other ground surface deformation) by examining SAR
backscatter change with data acquired by the 10-m Copernicus Sentinel-1 (S1) satellites [see full
description in Handwerger et al. (2022)]. We measured the change in SAR backscatter by using
the log ratio approach, defined as
$$I_{ratio} = 10 \times log_{10}(\frac{\sigma_{pre}{}^0}{\sigma_{post}{}^0}) \qquad\qquad (3)$$
where $\sigma_{pre}^0$ is a pre-event image stack (defined as the temporal median) of SAR backscatter and
$\sigma_{post}^0$ is a post-event image stack. Similar to the HazMapper method, our approach requires
selection of an event date, pre-event window (months), post-event window (months) and slope
threshold (°). No cloud-cover threshold is needed since SAR penetrates clouds. We used a 3 month
pre- and post-event window and 0° slope threshold to identify ground surface changes associated
with the debris flows. We then created a binary map to highlight debris flows by removing all
pixels with $I_{ratio} < 99$[th] percentile value [i.e., threshold suggested by Handwerger et al. (2022)].
Identified debris flow source areas and deposition sites were confirmed by field investigation (N.J.
Finnegan) and named after the locations where they deposited (i.e., Big Creek, Mill Creek, and
Nacimiento).

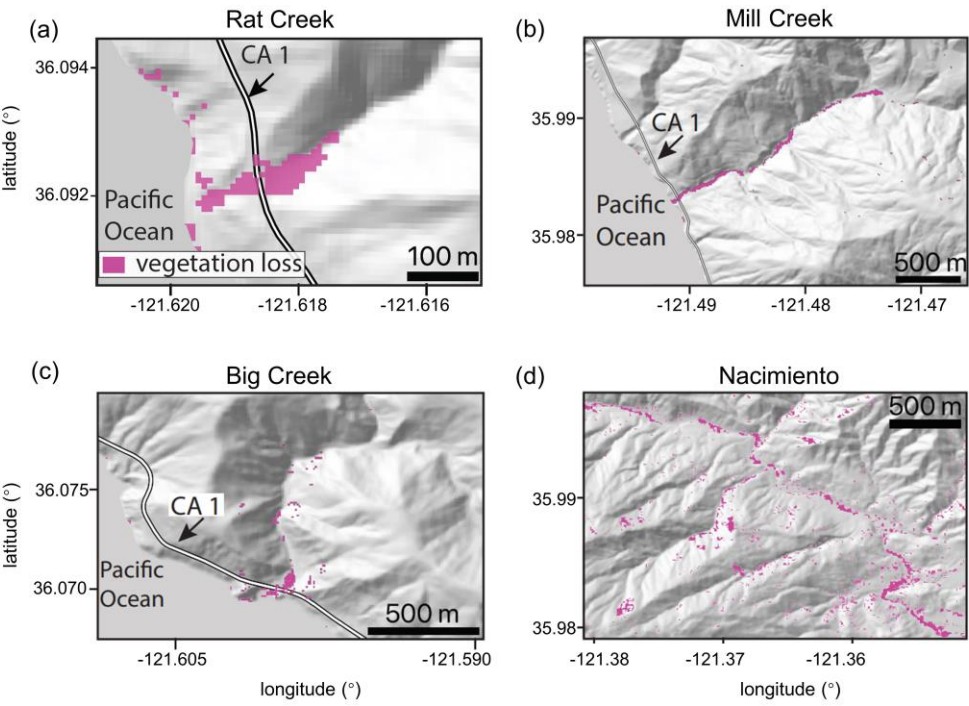

**Fig. 3|** Identified debris flow sites using rdNDVI vegetation change within the Dolan burn scar. We convert the rdNDVI data into a binary map by setting a threshold value, which yields only the likely debris flow locations and drape these maps over a topographic hillshade. (a)–(d) Sentinel-2 rdNDVI vegetation change at (a) Rat Creek, (b) Mill Creek, (c) Big Creek, and (d) the Nacimiento River.

## 2.2 Debris flow geologic setting

According to the USGS National Elevation Dataset 30-m digital elevation model (DEM), the Rat Creek debris flow sits at the base of a 1$^{st}$ order catchment with a drainage area of 2.23 km$^2$. Mill Creek, Big Creek, and Nacimiento debris flows were initiated within extremely steep, intensely burned, 1$^{st}$ order catchments, but were deposited in 2$^{nd}$, 3$^{rd}$, and 3$^{rd}$ Strahler stream order channels, respectively. All four debris flows were channelized. Rat Creek, Mill Creek, and Big Creek debris flow deposition sites have elevations ranging from 20–60 m, while Nacimiento debris flow deposited at an elevation of ~440 m above sea level. We calculate catchment slopes using the DEM and the slope calculation function in ArcMap. The average slope of the catchments containing Rat Creek and Mill Creek debris flow deposition sites is ~25°. The average catchment slope of Big Creek deposition site is ~28° and Nacimiento is ~21°. For debris flow source areas, the average and maximum slopes of Mill Creek are 23° and 39°, 21° and 43° for Big Creek, and 24° and 41°

for Nacimiento. According to the Soil Survey Geographic Database and California geologic map data, surface soils at the three coastal debris flow sites (i.e., Rat Creek, Mill Creek, and Big Creek) are texturally classified as loam with underlying Franciscan Complex sedimentary rocks of Jurassic to Cretaceous age. Soil at Nacimiento is classified as sandy loam with underlying Upper Cretaceous and Paleocene marine sedimentary rocks from the Dip Creek Formation, Asuncion Group, Shut-In Formation, Italian Flat Formation, Steve Creek Formation, and El Piojo Formation. Mill Creek, Big Creek, and Nacimiento were relatively large debris flows with runout lengths between ~2–5 km, while Rat Creek occurred in a smaller catchment and had a runout length of ~300 m. The difference in runout length and debris flow size is primarily controlled by upstream catchment size, however for the three coastal debris flow events at Rat Creek, Big Creek, and Mill Creek, also constrained by the downslope ocean. We note that there were likely more debris flows triggered during the AR event. The four debris flow events highlighted here were identified during brief post-event field excursions due to their intersection with major roadways. Given that our primary goal here is to demonstrate the utility of WRF-Hydro – a comprehensive catalogue of debris flows is beyond the scope of this study, although underway by other researchers (Cavagnaro et al., 2021).

## 3 WRF-Hydro

### 3.1 Model description

WRF-Hydro is an open-source physics-based community model that simulates land surface hydrologic processes. It includes the Noah-Multiparameterization (Noah-MP) land surface model (LSM; Niu et al., 2011), terrain routing module, channel routing module, and a conceptual baseflow bucket model. The Noah-MP LSM is a 1-dimensional column model that calculates vertical energy fluxes (i.e., sensible and latent heat, net radiation), moisture (i.e., canopy interception, infiltration, infiltration excess, deep percolation), and soil thermal and moisture states on the LSM grid (1 km in our application). The infiltration excess, ponded water depth, and soil moisture are then disaggregated using a time-step weighted method (Gochis & Chen, 2003) and sent to the terrain routing module which simulates subsurface and overland flows on a finer terrain routing grid (100 m in our application). According to the mass balance, local infiltration excess, overland flow, and exfiltration from baseflow contribute to the surface head which flows into river channels if defined retention depth is exceeded. The channel routing module then calculates channelized flows assuming a trapezoidal channel shape (Fig. B2). Parameters related to the trapezoidal channel, such as channel bottom width ($B_w$), Manning's roughness coefficient (n), and channel side slope (z) are functions of channel stream order (Fig. B3 and Table B1). Channelized streamflow is computed at spatial resolutions ranging from 1.5 m to 100 m depending on the channel stream order (Table B1). Computed streamflow is then output on the 100-m grid. Equations used to compute infiltration excess, overland flow, and channelized flow are provided in Sect. 3.3 and 3.4.

By default, WRF-Hydro uses Moderate Resolution Imaging Spectroradiometer (MODIS) Modified International Geosphere-Biosphere Program (IGBP) 20-category land cover product as land cover (Fig. B4) and 1-km Natural Resources Conservation Service State Soil Geographic (STATSGO) database for soil type classification (Fig. B5; Miller & White, 1998). Land surface properties including canopy height (HVT), maximum carboxylation rate (VCMX25), and overland flow roughness (OV_ROUGH2D) are functions of land cover type (Table B2 & Fig. B4). Default soil hydraulic parameters in WRF-Hydro (i.e., soil porosity, grain size distribution index, and saturated hydraulic conductivity) are based on Cosby et al.'s (1984) soil analysis (Table B3) and are used to map onto the STATSGO 16 soil texture types (Fig. B5).

## 3.2 Meteorological forcing files

WRF-Hydro is used in a standalone mode (i.e., it is not interactively coupled with the atmospheric component of WRF), but rather is forced with a combination of Phase 2 North American Land Data Assimilation System (NLDAS-2) meteorological data and MRMS radar-only quantitative precipitation (Zhang et al., 2011, 2014, 2016). A description of the MRMS dataset and uncertainties therein can be found in Appendix A. NLDAS-2 provides hourly forcing data including incoming shortwave and longwave radiation, 2-m specific humidity and air temperature, surface pressure, and 10-m wind speed at 1/8-degree spatial resolution. MRMS provides hourly precipitation rates at 1-km resolution.

## 3.3 Overland flow routing and output

The Noah-MP LSM calculates rate of infiltration excess following Chen & Dudhia (2001):

$$\frac{\partial h}{\partial t} = \frac{\partial P_d}{\partial t}\left\{1 - \frac{\left[\sum_{i=1}^{4}\Delta D_i(\theta_s-\theta_i)\right]\left[1-exp\left(-k\frac{K_s}{K_{ref}}\delta_t\right)\right]}{P_d+\left[\sum_{i=1}^{4}\Delta D_i(\theta_s-\theta_i)\right]\left[1-exp\left(-k\frac{K_s}{K_{ref}}\delta_t\right)\right]}\right\}$$  (4)

where $h$ (m) is the surface water depth and $t$ is the time. $P_d$ (m) is the precipitation not intercepted by the canopy; $\Delta D_i$ (m) is the depth of soil layer i; $\theta_i$ is the soil moisture in soil layer i; $\theta_s$ is the soil porosity; $K_s$ (m s$^{-1}$) is the saturated hydraulic conductivity; $K_{ref}$ is $2 \times 10^{-6}$ m s$^{-1}$ which represents the saturated hydraulic conductivity of the silty–clay–loam soil texture chosen as a reference; $\delta_t$ (s) is the model time step; and $k$ which is equal to 3.0 is the runoff–infiltration partitioning parameter [the same as $kdt_{ref}$ in Chen & Dudhia (2001)].

Noah-MP passes excess water to the terrain routing module, which simulates overland flow using
a 2-dimensional fully-unsteady, explicit, finite-difference diffusive wave equation adapted from
Julien et al. (1995) and Ogden (1997). In this application, overland flow is computed at each 6
second time step and is archived hourly at 100-m spatial resolution. The diffusive wave equation
is considered improved compared to the traditionally used kinematic wave formulation in that it
accounts for backwater effects and flow over adverse slopes. The diffusive wave formulation is
the simplified form of the Saint Venant equations, i.e., continuity and momentum equations for a
shallow water wave. The 2-dimensional continuity equation for a flood wave is:
$$\frac{\partial h}{\partial t} + \frac{\partial q_x}{\partial x} + \frac{\partial q_y}{\partial y} = i_e \tag{5}$$

where $h$ is the surface flow depth, $q_x$ and $q_y$ are the unit discharges in the x- and y-directions,
respectively, and $i_e$ is the infiltration excess. Manning's equation which considers momentum loss
is used to calculate $q$. In the x-direction:
$$q_x = \alpha_x h^\beta \tag{6}$$

Where $\beta$ is a unit dependent coefficient equal to $\frac{5}{3}$, and
$$\alpha_x = \frac{S_{fx}^{1/2}}{n_{ov}} \tag{7}$$

where $n_{ov}$ is the tunable overland flow roughness coefficient. The momentum equation in the x-
direction is given by:
$$S_{fx} = S_{ox} - \frac{\partial h}{\partial x} \tag{8}$$

where $S_{fx}$ is the friction slope, $S_{ox}$ is the terrain slope, and $\frac{\partial h}{\partial x}$ is the change in surface flow depth
in the x-direction.
Off-the-shelf, WRF-Hydro does not output overland flow at terrain routing grids (100 m), however
it is computed in the background to determine channelized streamflow. One key advance made in
this work is that we modified WRF-Hydro source code to output overland flow (see the Code
availability statement for the modified source code). Overland flow depth (m) was converted to
overland discharge (m$^3$ s$^{-1}$) by multiplying flow depth by grid cell area (10,000 m$^2$) and dividing
by the LSM time step (1 h).

## 3.4 Channel routing

If overland flow intersects grid cells identified as channel grids (2$^{nd}$ Strahler stream order and
above; pre-defined by the hydrologically conditioned USGS 30-m DEM), the channel routing
module routes the water as channelized streamflow using a 1-dimensional, explicit, variable time-
stepping diffusive wave formulation. In this work, the channel routing module calculates

streamflow at 6-s temporal resolution and spatial resolutions ranging from 1.5 m to 100 m depending on the channel stream order (Fig. B3 and Table B1). Similarly, the continuity equation for channel routing is given as:

$$\frac{\partial A}{\partial t} + \frac{\partial Q}{\partial s} = q_l \tag{9}$$

and the momentum equation is given as:

$$\frac{\partial Q}{\partial t} + \frac{\partial (\frac{\gamma Q^2}{A})}{\partial s} + gA\,\frac{\partial H}{\partial s} = -gAS_f \tag{10}$$

where $s$ is the streamwise coordinate, $H$ is water surface elevation, $A$ is the flow cross-sectional area calculated as $(B_w + H\,z)H$ (Fig. B2), $q_l$ is the lateral inflow rate into the channel grid, $Q$ is the flow rate, $\gamma$ is a momentum correction factor, $g$ is acceleration due to gravity, and $S_f$ is the friction slope computed as:

$$S_f = (\frac{Q}{K})^2 \tag{11}$$

where $K$ is the conveyance computed from the Manning's equation:

$$K = \frac{C_m}{n} AR^{2/3} \tag{12}$$

where $n$ is the Manning's roughness coefficient, $A$ is the channel cross-sectional area, $R$ is the hydraulic radius $(A/P)$, $P$ is the wetted perimeter, and $C_m$ is a dimensional constant (1.486 for English units or 1.0 for SI units).

## 4 Model simulation, calibration, and validation

### 4.1 Model domain

The WRF-Hydro model domain spans regions in California including the Coast Ranges, Monterey Bay, and the Central Valley, and covers several burn scars from the 2020 wildfire season (Fig. 1a). Here we focus our analysis on the Dolan burn scar where the hazardous debris flows occurred (Fig. 1b).

To calibrate and validate WRF-Hydro output, we use soil moisture observations from two Physical Sciences Laboratory (PSL) monitoring stations [i.e., Lockwood (lwd) and Gilroy (gry)] (Fig. 1a). Due to the Mediterranean climate of California, many USGS stream gages experience low or no flow during the dry season. In addition, many gages are under manual regulation to mitigate wet-season flood risks and better distribute water resources. As such, it can be challenging to obtain natural streamflow observations for model calibration. Here, three USGS stream gages [i.e., Arroyo Seco NR Greenfield, CA (ID 11151870), Arroyo Seco NR Soledad, CA (ID 11152000), and Arroyo Seco BL Reliz C NR Soledad, CA (ID 11152050)] (Fig. 1a) on streams that have measurable flows during our study period and are free of human regulation are used. These gages

are located downstream of the Dolan burn scar and hence are useful in calibrating the parameters associated with burn scar effects. The PSL soil moisture observations were recorded at 2-minute intervals and USGS streamflow gage data were recorded at 15-minute intervals, but we perform all observation-model comparisons at hourly-mean resolution.

## 4.2 Baseline simulation and soil moisture calibration

WRF-Hydro was initialized with National Centers for Environmental Prediction (NCEP) FNL (Final) Operational Global Analysis data and was run from January 1–31, 2021. We performed the baseline simulation by modifying WRF-Hydro default parameters (Table B3) based on a calibration using soil moisture observations from stations lwd and gry. Neither PSL station is located in a burn scar. Since the baseline simulation includes no postfire characteristics, it can also be regarded as the "pre-fire" scenario. Soil moisture at 10 cm below ground in the baseline simulation was calibrated by performing a domain-wide adjustment of soil porosity and grain size distribution index at the simulation start (Table B3). We then allowed the model to spin up from January 1–10 before using January 11–31 for validation. Using a relatively short spin-up period is justified because prior to the AR event, little rain fell on the Dolan burn scar (i.e., ~400 mm of rainfall fell from June to December 2020). As such, in the months preceding the debris flow events, soil moisture observations indicate already dry conditions prior to our 10 day spin up.

After calibration, the simulated soil moisture closely mimics ground-based PSL observations (Fig. 4). Both the observed magnitude and variability are well captured, with the simulated ±1 standard deviation envelope largely encompassing PSL observations during the AR. Model performance was evaluated using four quantitative metrics, i.e., correlation coefficient ($r$), root mean square error (RMSE), mean absolute error (MAE), and Kling-Gupta efficiency (KGE; Gupta et al., 2009; Kling et al., 2012). KGE has previously been used in soil moisture calibration applications (e.g., Lahmers et al., 2019; Vergopolan et al., 2020) and is computed as follows:

$$KGE = 1 - \sqrt{(r - 1)^2 + (\alpha - 1)^2 + (\beta - 1)^2} \tag{13}$$

where $r$ is the correlation coefficient between the observation and simulation, $\alpha$ is the ratio of the standard deviation of simulation to the standard deviation of observation, and $\beta$ is the ratio of the mean of simulation to the mean of observation. KGEs close to 1 indicate a high-level consistency between the simulation and observation, while negative KGEs indicate poor model performance (Andersson et al., 2017; Schönfelder et al., 2017).

The model's ability to simulate soil moisture substantially improves after calibration (Fig. 4; Table 1). KGE values approach 1 (0.72 at lwd and 0.88 at gry), indicating that WRF-Hydro adequately simulates the hydrologic environment and its response to meteorological changes.

430

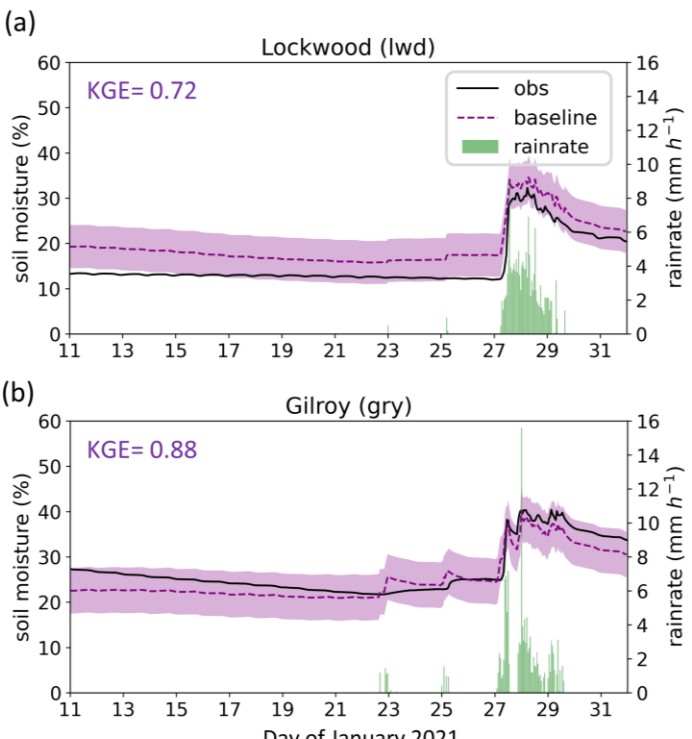

431

**Fig. 4|** Precipitation, observed and simulated soil moisture at two PSL soil moisture stations.
January 11–31, 2021 MRMS precipitation (mm h$^{-1}$; green bars) and observed (%; black line) and
simulated volumetric soil moisture 10 cm below ground in the baseline simulation (%; purple
dashed line) at PSL sites (a) Lockwood (lwd) and (b) Gilroy (gry). Envelope of purple shading
depicts ±1 standard deviation of model simulated soil moisture. KGE scores are provided at top
left for each station.

*Table 1*

*Evaluation metrics of simulated soil moisture and streamflow*

| | **Soil moisture (Default / Baseline)** | | | |
|---|---|---|---|---|
| **Station** | ***r*** | **RMSE** | **MAE** | **KGE** |
| **lwd** | 0.97 / 0.98 | 7.06 / 4.32 | 5.21 / 4.16 | 0.10 / 0.72 |

| gry | 0.94 / 0.94 | 5.19 / 2.53 | 11.12 / 2.31 | 0.80 / 0.88 |
|---|---|---|---|---|
| **Streamflow (Baseline / Burn scar)** | | | | |
| **Station** | *r* | **RMSE** | **MAE** | **NSE** |
| **1870** | 0.28 / 0.93 | 39.29 / 14.69 | 16.05 / 6.14 | -0.17 / 0.84 |
| **2000** | 0.26 / 0.86 | 51.22 / 24.92 | 20.11 / 10.00 | -0.15 / 0.73 |
| **2050** | 0.25 / 0.81 | 49.96 / 27.43 | 19.64 / 11.65 | -0.38 / 0.53 |

**Table 1|** Quantitative evaluation metrics for the simulated soil moisture and streamflow when compared against observations. The metrics include the Pearson correlation coefficient (*r*), root mean square error (RMSE), and mean absolute error (MAE). In addition, the comprehensive metrics Kling-Gupta efficiency (KGE) and Nash-Sutcliffe efficiency (NSE) are used to evaluate model-simulated soil moisture and streamflow, respectively. For soil moisture, the numbers in front of "/" are calculated between the default run (i.e., uncalibrated run) and the observations, whereas the numbers following "/" are the corresponding values in the baseline simulation (the purple dashed line in Fig. 4). For streamflow, the numbers in front of "/" are computed between the baseline run (purple dashed line in Fig. 6) and the observations, while the numbers behind "/" are for burn scar simulation (red line in Fig. 6). If the model performance regarding a certain metric is enhanced in the burn scar simulation, the number after "/" is underlined.

## 4.3 Burn scar simulation and streamflow calibration

To simulate effects of wildfire burn scars on hydrologic processes and debris flow susceptibility, we made two modifications to the baseline simulation soil moisture calibrated model configuration. First, we changed the land cover type within the burn scar perimeter to its nearest LSM analogue, i.e., "barren and sparsely vegetated". The switch to barren land causes: (1) height of the canopy (HVT) to decrease to 0.5 m; (2) maximum rate of carboxylation at 25°C (VCMX25) to decrease to 0 $\mu mol\ CO_2/(m^2 \cdot s)$; and (3) overland flow roughness coefficient (OV_ROUGH2D) to decrease to 0.035 (Fig. 5a–c) from default values (Fig. B4 and Table B2).

The second adjustment was to decrease soil infiltration rates within the burn scar perimeter, achieved by reducing soil saturated hydraulic conductivity (DKSAT; Fig. 5d; Scott & van Wyk, 1990; Cerdà, 1998; Robichaud, 2000; Martin & Moody, 2001) from default values (Table B3). Consistent with the hydrophobicity of burned soils, we calibrate the burn scar simulation by systematically exploring a range of burn scar area saturated hydraulic conductivities (0 to $3 \times 10^{-7}$ m s$^{-1}$ with a $5 \times 10^{-8}$ m s$^{-1}$ increment), with the goal of reproducing streamflow behavior similar to USGS gage observations. We found that a value of $1.5 \times 10^{-7}$ m s$^{-1}$ gives the highest Nash-Sutcliffe efficiency (NSE; Nash & Sutcliffe, 1970) across all three USGS stream gages (Table 1). NSE and KGE are the two most widely used metrics for calibration and evaluation of hydrologic models.

The NSE has previously been used in streamflow calibration applications (e.g., Xia et al., 2012;
Bitew & Gebremichael, 2011), and it is calculated as follows:

$$NSE = 1 - \frac{\sum_{t=1}^{t=T}(Q_{sim}(t)-Q_{obs}(t))^2}{\sum_{t=1}^{t=T}(Q_{obs}(t)-\overline{Q_{obs}})^2} \tag{14}$$


where $T$ is the length of the time series, $Q_{sim}(t)$ and $Q_{obs}(t)$ are the simulated and observed
discharge at time $t$, respectively, and $\overline{Q_{obs}}$ is the mean observed discharge. By definition, NSEs of
1 indicate perfect correspondence between the simulated and observed streamflow. Positive NSEs
indicate that the model streamflow has a greater explanatory power than the mean of the
observations, whereas negative NSEs represent poor model performance (Moriasi et al., 2007;
Schaefli & Gupta, 2007). When burn scar characteristics are included, evaluation metrics including
$r$, RMSE, and MAE all improve, while NSEs increase from negative values in the baseline to 0.84,
0.73, and 0.53 at gages 1870, 2000, and 2050, respectively. Higher correlation and NSE scores
and lower errors indicate the above mentioned burn scar parameter changes improve the model's
ability to simulate streamflow observations downstream of the burn scar (Table 1).

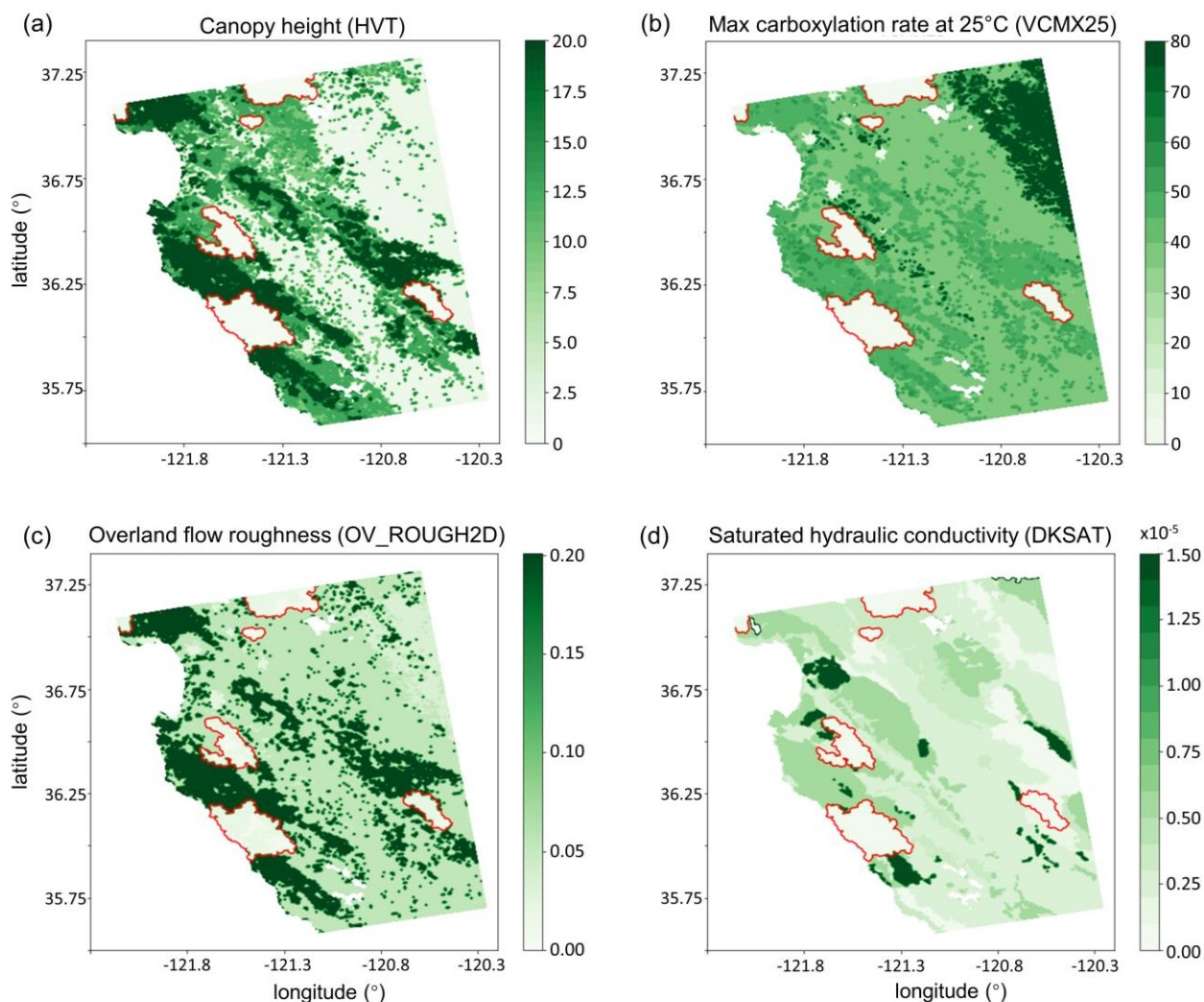

**Fig. 5|** Parameter setting in the WRF-Hydro burn scar simulation. (a) The height of the canopy
(HVT; m; shading), (b) maximum rate of carboxylation at 25°C (VCMX25; $\mu mol\ CO_2/(m^2 \cdot s)$;
shading), (c) overland flow roughness coefficient (OV_ROUGH2D; shading), and (d) saturated
hydraulic conductivity (DKSAT; m s⁻¹; shading) in the burn scar simulation.

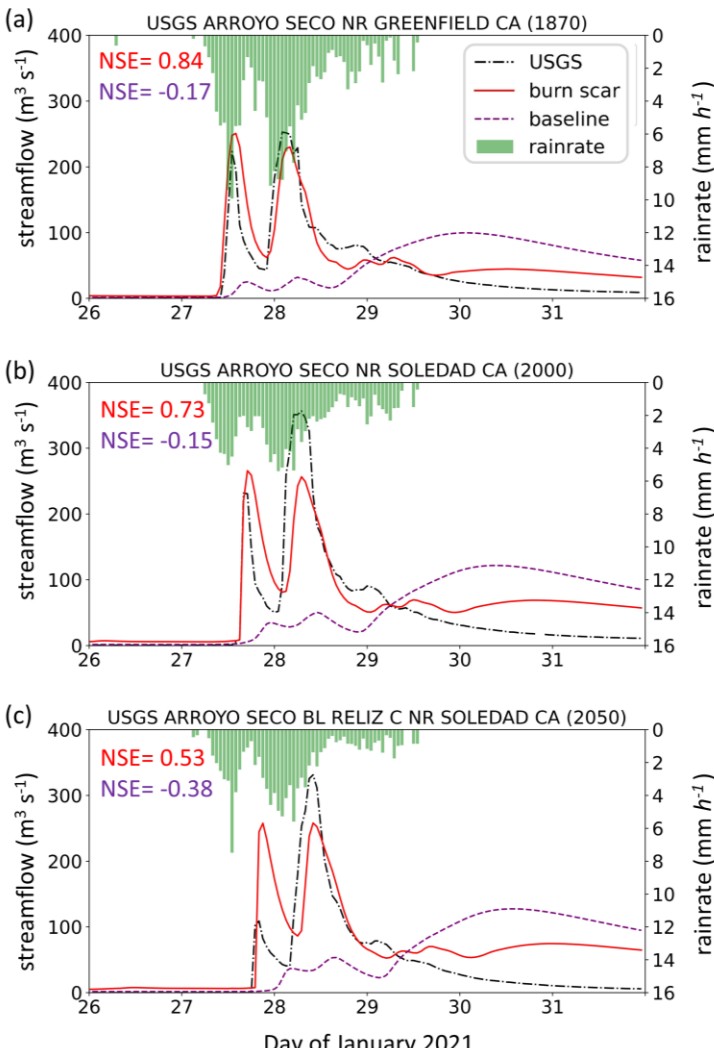

**Fig. 6|** Precipitation, observed and simulated streamflow at three USGS stream gages. January 26–31, 2021 MRMS precipitation (mm h$^{-1}$; green bars), observed (m$^3$ s$^{-1}$; black dash dotted line) and simulated streamflow in baseline simulation (m$^3$ s$^{-1}$; purple dashed line) and burn scar simulation (m$^3$ s$^{-1}$; red line) at (a) Arroyo Seco NR Greenfield, CA (ID 11151870), (b) Arroyo Seco NR Soledad, CA (ID 11152000), and (c) Arroyo Seco BL Reliz C NR Soledad, CA (ID 11152050). NSE scores for baseline (purple) and burn scar simulations (red) are shown at top left.

## 5 Results

### 5.1 Hydrologic response due to burn scar incorporation

The pre-fire baseline simulation fails to capture the hydrologic behavior observed at the USGS gages located within the burn scar (Fig. 6). Incorporation of burn scar characteristics substantially alters the hydrologic response of the model and provides much higher fidelity streamflow simulations (Fig. 6). Observed hydrographs are characterized by two early streamflow peaks related to two precipitation bursts on January 27[th] and 28[th]. Our burn scar simulation captures this behavior, while the baseline simulation streamflow peaks just once, with a lower magnitude and an ~3-day lag after peak precipitation (Fig. 6). The steep rising limbs and high magnitude discharge peaks of the burn scar hydrograph are indicative of flash flooding. Compared with the pre-fire baseline scenario, the burn scar's barren land and low infiltration rate substantially accelerate drainage rates and increase discharge volume into stream channels.

### 5.2 Hydrologic response at four debris flow sites

Mill Creek, Big Creek, and Nacimiento deposits are located in channels of 2[nd] Strahler stream order or above so they are simulated as channelized streamflow in our WRF-Hydro simulations. Due to its low stream order (1[st] Strahler stream order), Rat Creek is modeled entirely as overland flow in our WRF-Hydro simulations. At the four debris flow sites, we use three metrics to characterize hydrologic anomalies: (1) accumulated runoff volume, (2) peak discharge, and (3) time to peak discharge. Fig. 7 depicts accumulated channelized discharge volume (blue shading) and accumulated overland discharge volume (yellow-red shading) from January 27[th] 00:00 to 28[th] 12:00 near the four debris flow sites in the burn scar simulation. Accumulation time period is chosen such that it covers the first two runoff surges in the simulated hydrographs which are likely associated with debris flows (Fig. 8) given that nearly concurrent peak rainfall intensity and peak discharge is a signature characteristic of debris flows (Kean et al., 2011). Runoff volume is on the order of $10^4$ m$^3$ at Rat Creek and $10^6$ m$^3$ at the other three sites.

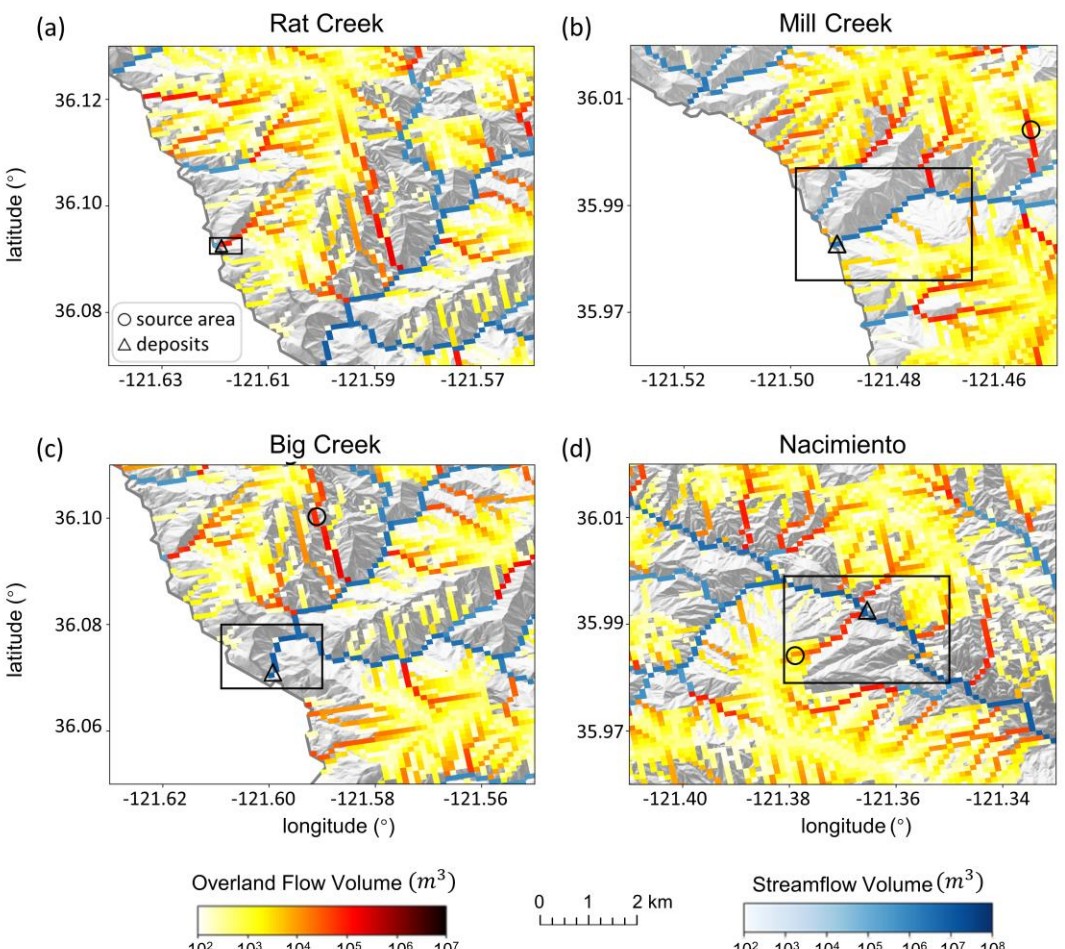

## Simulated overland flow and streamflow in burn scar simulation

**Fig. 7|** WRF-Hydro simulated overland flow and streamflow in the burn scar simulation. (a)–(d)
Total volume of accumulated overland flow (m$^3$; yellow-red shading) and streamflow (m$^3$; blue
shading) between January 27$^{th}$ 00:00 and 28$^{th}$ 12:00 at four debris flow sites draped over a hillshade
of topography. Black rectangles correspond to domains in Fig. 3a–d. Black circles and triangles
indicate debris flow source areas and deposits, respectively.

Dramatic hydrographic changes after inclusion of burn scar characteristics are simulated at debris
flow source areas (Fig. B6 and Table B4) and deposition sites (Fig. 8 and Table 2). Here, to
emphasize the high susceptibility downstream, our analysis is focused on debris flow deposits. At
Rat Creek, where a section of CA1 collapsed, the magnitude of discharge substantially increases,
and overland flow surges are concurrent with rainfall bursts (Fig. 8a). Total discharge accumulated
during the AR event increases approximately eight-fold (791%), and peak discharge more than

triples compared to the baseline simulation (Fig. 8a and Table 2). At Mill Creek, Big Creek, and Nacimiento, baseline hydrographs are characterized by less variability, muted responses to two early precipitation bursts, and a delayed third discharge peak that does not occur until ~3 days after AR passage (Fig. 8b–d). Maximum discharge peaks in the baseline hydrographs lag those in the burn scar simulation by ~2 days (Fig. 8b–d; Table 2). In the burn scar simulation, total volume substantially increases at the three channelized sites – total volume increases ~650% at Mill Creek, ~891% at Big Creek, and ~829% at Nacimiento (Fig. 8b–d and Table 2), and the absolute increase in volume is on the order of $10^6$ m$^3$ (Table 2). Peak discharge more than triples at Mill Creek and Big Creek and more than quadruples at Nacimiento. Additionally, response times of the peak in discharge to the peak in precipitation decrease to less than an hour, highlighting the simulated flashiness of the burned catchments.

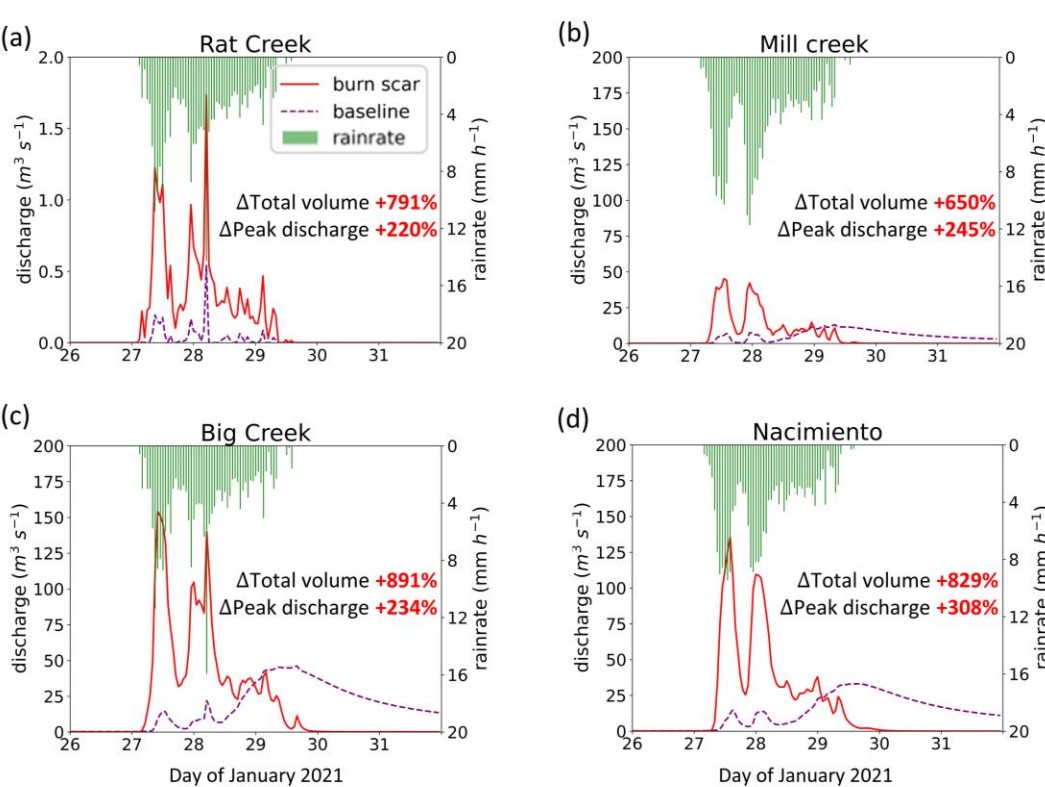

**Fig. 8**| WRF-Hydro simulated discharge time-series at four debris flow deposition locations. (a)–(d) MRMS precipitation (mm h$^{-1}$; green bars) and simulated discharge time-series for January 26[th] 00:00 to 31[st] 23:00 at (a) Rat Creek, (b) Mill Creek, (c) Big Creek, and (d) Nacimiento deposition locations (black triangles in Fig. 7a–d) in baseline simulation (m$^3$ s$^{-1}$; purple dashed line) and burn scar simulation (m$^3$ s$^{-1}$; red line).

563

564

*Table 2*

*The total runoff volume, peak discharge, and peak timing at debris-flow deposits*

| Site name | Baseline simulation | | | Burn scar simulation | | | |
|---|---|---|---|---|---|---|---|
| | Total volume (m³) | Peak discharge (m³ s⁻¹) | Highest peak timing | Total volume (m³) | Peak discharge (m³ s⁻¹) | 1st Peak timing | 2nd Peak timing |
| **Rat Creek** | 6,897 | 0.54 | 28th 05:00 | 61,425 (+791%) | 1.73 (+220%) | 27th 09:00 | 28th 05:00 |
| **Mill Creek** | 312,925 | 13.10 | 29th 08:00 | 2,347,457 (+650%) | 45.21 (+245%) | 27th 13:00 | 27th 23:00 |
| **Big Creek** | 842,808 | 46.10 | 29th 16:00 | 8,354,095 (+891%) | 154.10 (+234%) | 27th 10:00 | 28th 05:00 |
| **Nacimiento** | 743,531 | 33.15 | 29th 16:00 | 6,904,706 (+829%) | 135.41 (+308%) | 27th 14:00 | 28th 00:00 |

567

**Table 2|** The total runoff volume, peak discharge, and peak timing in the baseline and burn scar simulations from January 27th 00:00 to 31st 23:00 at deposition sites of Rat Creek, Mill Creek, Big Creek, and Nacimiento debris flows (black triangles in Fig. 7a–d). The peak timing shown in the baseline simulation is for the highest peak. The percent change of the total volume and peak discharge in the burn scar simulation relative to the baseline simulation are shown in parentheses.

573

574

**5.3 Debris flow susceptibility assessment for the Dolan burn scar**

Since high magnitude runoff is often the cause and precursor of runoff-generated debris flows in burned areas (Cannon et al., 2003, 2008; Rengers et al., 2016), we use peak discharge of overland flow and streamflow to assess runoff-generated debris flow susceptibility under pre-fire (i.e., baseline; Fig. 9a&d) and postfire (i.e., burn scar simulation; Fig. 9b&e) conditions [we conduct similar analyses using accumulated discharge volume in Figs. B7–9 and Table B5 in Appendix B]. We assess changes at both stream and catchment levels and use the difference between burn scar and baseline simulations to assess added debris flow susceptibility (Fig. 9c&f). Consistent with the increasing erosive and entrainment power associated with increasing discharge, our debris flow susceptibility increases as peak discharge increases. To reduce the effects of catchment size on the peak discharge-based susceptibility levels, we normalize a catchment's discharge by the area of the catchment (Leopold et al., 1964; McCormick et al., 2009; Fig. 9d–f). Non-normalized catchment susceptibility maps are also provided (Fig. B10).

588

In the pre-fire baseline simulation, the AR-induced precipitation produces lower debris flow susceptibility over most of the domain, but elevated susceptibility along stream channels (Fig. 9a). We note no substantial differences between areas in or out of the burn scar. In the burn scar simulation, debris flow susceptibility levels increase across the Dolan burn scar and along channels outside but downstream of the burn scar (Fig. 9b–c). The peak discharge near Rat Creek, Big Creek, Mill Creek, and Nacimiento more than triples (Table 2 & Fig. 9a–c). Within the burn scar, susceptibility along major stream channels, such as the Nacimiento River and San Antonio River increase. Outside the burn scar, susceptibility levels along river channels downstream of the burn scar, such as the Arroyo Seco River, also increase (Fig. 9c).

At the catchment level, debris flow susceptibility is assessed using peak discharge normalized by catchment areas at the outlet of each catchment between January 27th 00:00 to 28th 12:00 (Fig. 9d–f). The catchment-area normalized peak discharge is classified into five categories based on equal intervals on $\log_{10}$ scale. The susceptibility categorization follows: "very low" ($\sim 10^{-2}$ m$^3$ s$^{-1}$ km$^{-2}$), "low" ($\sim 10^{-1}$ m$^3$ s$^{-1}$ km$^{-2}$), "medium" ($\sim 10^{0}$ m$^3$ s$^{-1}$ km$^{-2}$), "high" ($\sim 10^{1}$ m$^3$ s$^{-1}$ km$^{-2}$), and "very high" ($\sim 10^{2}$ m$^3$ s$^{-1}$ km$^{-2}$). In the baseline simulation, majority of catchments are subject to low or very low debris flow susceptibility with normalized peak discharge less than 1 m$^3$ s$^{-1}$ km$^{-2}$ (Fig. 9d). In the burn scar simulation, about half of the catchments within the Dolan burn scar have medium susceptibility or above, and about 1/4 of basins are subject to high to very high debris flow susceptibility (Fig. 9e and Table 3). The additional debris flow susceptibility brought about by the inclusion of wildfire burn scar characteristics is substantial (Fig. 9f).

To summarize changes in debris flow susceptibility as a result of including burn scar characteristics in WRF-Hydro simulations, we create distributions of pre-fire baseline and burn scar catchment-area normalized peak discharge from the 404 catchments located within the Dolan burn scar perimeter (Fig. 10). After incorporating burn scar characteristics, the full distribution shifts to the right, indicating increased susceptibility levels – a shift considered robust by a Student's t-test (*p* value: 5.3E-23). A quantitative assessment of this shift indicates that both the mean and the standard deviation of catchment area normalized peak discharge increase by more than 300% (Table 3). We also assess shifts at a range of distribution percentiles: 5P: 375%, 25P: 500%, 50P: 447%, 75P: 341%, and 95P: 366% (Table 3). In the burn scar simulation, more than half of catchments have normalized peak discharge > $10^{0}$ m$^3$ s$^{-1}$ km$^{-2}$ (i.e., medium susceptibility) and about 1/4 of catchments have normalized peak discharge > $10^{1}$ m$^3$ s$^{-1}$ km$^{-2}$ (i.e., high susceptibility) – values that correspond to the 70P and 90P of the baseline simulation, respectively. Disproportionate shifting of the distribution suggests that debris flow susceptibility increases non-linearly under simulated burn scar conditions.

Our catchment-area normalized peak discharge-based susceptibility assessment also indicates that the catchments containing Mill Creek, Big Creek, and Nacimiento have high or very high susceptibility (Fig. 9d–f), consistent with our (limited) debris flow observations. Other areas with

elevated susceptibility include catchments containing the Arroyo Seco and San Antonio Rivers. Beyond the burn scar perimeter, effects of fire expand to adjacent and downstream catchments, and some drainage basins along the Arroyo Seco and Nacimiento Rivers are simulated to have very high susceptibility, i.e., normalized peak discharge exceeds $10^2$ m$^3$ s$^{-1}$ km$^{-2}$ (Fig. 9e&f).

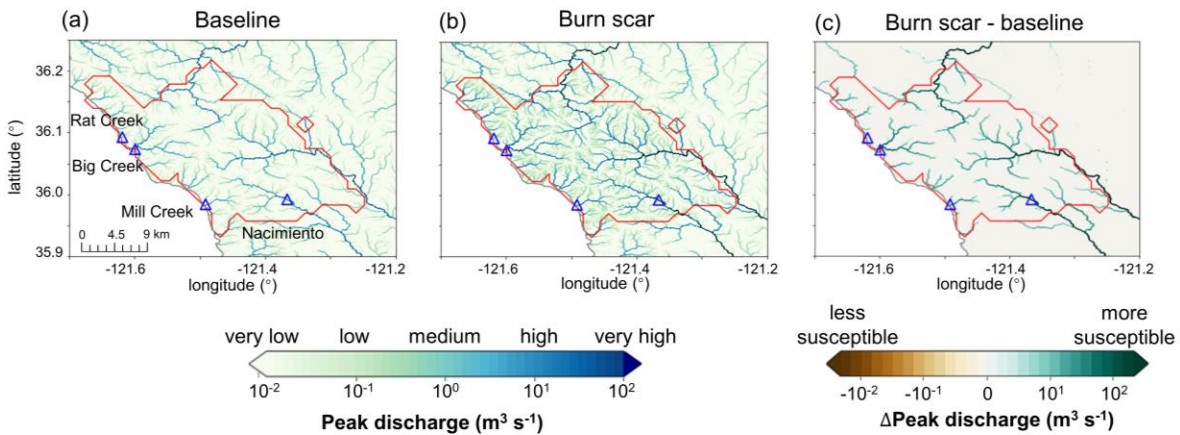

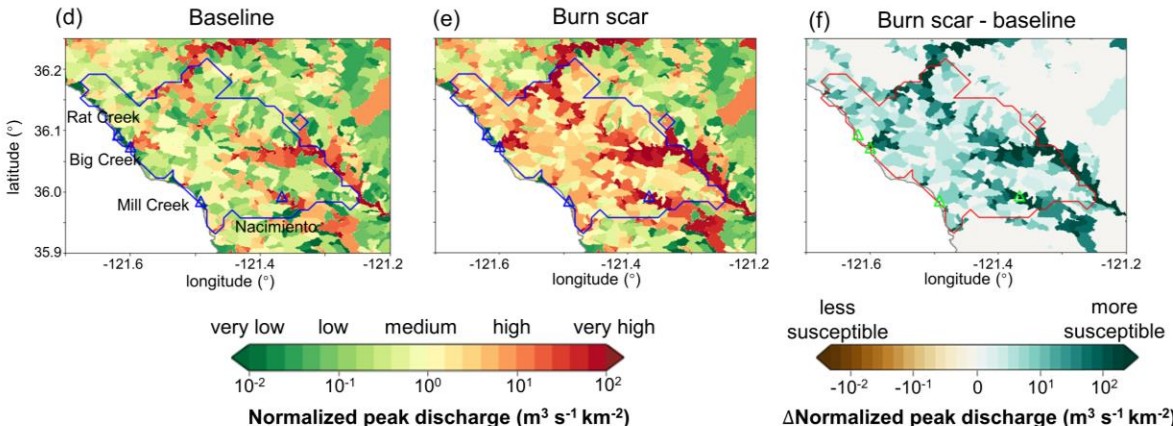

**Fig. 9|** Peak discharge-based postfire debris flow susceptibility. Peak discharge at individual stream level for the (a) baseline, (b) burn scar, and (c) difference between burn scar and baseline simulations from January 27th 00:00 to 28th 12:00 (m$^3$ s$^{-1}$). (d)–(f) Normalized peak discharge by catchment area at catchment level (m$^3$ s$^{-1}$ km$^{-2}$; shading). For each catchment, the peak discharge is the maximum discharge rate at the catchment outlet from January 27th 00:00 to 28th 12:00 divided by catchment area. Triangles stand for debris flow deposition locations and are annotated in (a) and (d). We conduct similar analyses using accumulated discharge volume in Fig. B7 in Appendix B.

*Table 3*
*Statistics of catchment area-normalized peak discharge in baseline and burn scar simulations*

|  | mean | std | 5P | 25P | 50P | 75P | 95P |
|---|---|---|---|---|---|---|---|
| **Baseline simulation** ($m^3 s^{-1} km^{-2}$) | 25.88 | $\pm 95.71$ | 0.04 | 0.14 | 0.76 | 8.21 | 129.54 |
| **Burn scar simulation** ($m^3 s^{-1} km^{-2}$) | 110.80 | $\pm 423.82$ | 0.19 | 0.84 | 4.16 | 36.21 | 603.15 |
| **Relative percent change** | 328% | 343% | 375% | 500% | 447% | 341% | 366% |

**Table 3|** Statistics, including the mean, standard deviation (std), 5P, 25P, 50P, 75P, and 95P, of
the catchment-area normalized peak discharge for all the 404 basins within the Dolan burn scar in
the baseline and burn scar simulation and their relative percent changes. We conduct similar
analyses using accumulated discharge volume in Table B5 in Appendix B.

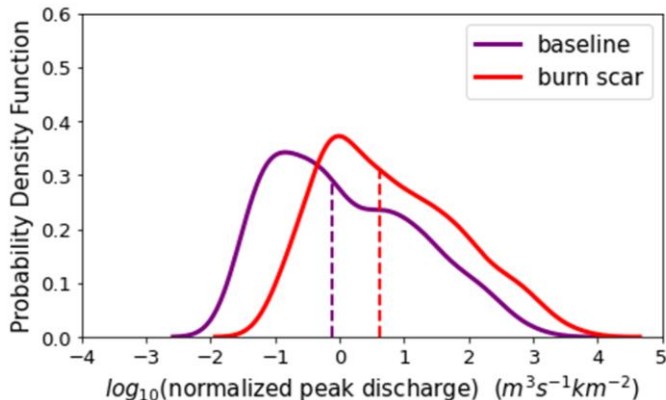


**Fig. 10|** Distributions of peak discharge at the outlet of the 404 catchments normalized by upstream
catchment areas within Dolan burn scar in the baseline simulation (purple line) and in the burn
scar simulation (red line). Dashed vertical lines indicate median values. We conduct similar
analyses using accumulated discharge volume in Fig. B8 in Appendix B.

## 6 Discussion

Given the historic and growing frequency of wildfires in the western U.S. (Williams et al., 2019;
Goss et al., 2020; Swain 2021) and globally (Flannigan et al., 2013; Jolly et al., 2015), developing
tools to investigate, better understand, and potentially predict changes in burn scar hydrology and
natural hazards at regional scales is critical. Here, we demonstrate the first use of WRF-Hydro to
simulate the susceptibility of a burn scar to postfire debris flows during a landfalling AR. We
augmented the default version of WRF-Hydro to output overland flow and to replicate burn scar
behavior by adjusting vegetation type and infiltration rate parameters. WRF-Hydro simulations
were validated against PSL soil moisture and USGS streamflow observations before we used
simulated peak discharge of streamflow and overland flow to characterize debris flow
susceptibility. A comparison between baseline and burn scar simulations demonstrated that
changes in hydraulic properties of burned areas causes drastic changes in surface flows, including
faster discharge response times, and greater peak discharge and total volumes, consistent with
findings from previous postfire hydrology studies (Anderson et al., 1976; Scott, 1993; Meixner &
Wohlgemuth, 2003; Kean et al., 2011; Kinoshita & Hogue, 2015; Brunkal & Santi, 2016; Williams
et al., 2022). At the catchment level, for the 404 catchments located within the Dolan burn scar,
median catchment area-normalized peak discharge increases by ~450% relative to the baseline. In
addition, Mill Creek, Big Creek, and Nacimiento basins were simulated to have high-to-very high
debris flow susceptibility, corresponding well with identified debris flow occurrences.

Despite methodological differences, our debris flow susceptibility map for this AR event is
generally consistent with the USGS' postfire, pre-AR, design-storm-based preliminary hazard
assessment (USGS, 2020). As described above, USGS preliminary hazard assessments use logistic
regression models to estimate the likelihood of debris flow occurrence and multivariate linear
regression models to estimate debris flow volumes. The USGS empirical approach is trained on
historical western U.S. debris flow occurrence and magnitude data and incorporates burn scar soil
erodibility and burn severity data (Cannon et al., 2010; Gartner et al., 2014; Staley et al., 2016).
For precipitation, the USGS assessment utilizes a design storm approach that assumes 1–5 year
return interval magnitude precipitation falls uniformly over a region/burn scar (USGS, 2020). For
the Dolan burn scar, both assessments find that large stream channels had relatively higher
susceptibility than small streams or overland areas. However, a close comparison of the two maps
reveals differences in spatial distribution of hazardous catchments. In the USGS assessment,
higher likelihood is predicted north and southeast of the burn scar, whereas in our assessment the
highest susceptibility occurs along major stream channels. We hypothesize that USGS-assessed
areas of higher hazard potential are related to their use of spatially uniform design-storm
precipitation (see Fig. 2 for MRMS precipitation footprint) and burn severity data (Burned Area
Emergency Response, 2020).

Comparison with the USGS hazard assessment framework suggests room for improvement in
WRF-Hydro-based assessments (i.e., inclusion of burn severity and soil erodibility data), but also
highlights the potential utility of working with spatially-distributed and time-varying precipitation.
However, this also means the accuracy of WRF-Hydro predictions depends on the accuracy of
precipitation forcing, and in our hindcast application, MRMS precipitation data (Appendix A).
Accordingly, our WRF-Hydro-based assessment could benefit from precipitation products
mosaiced from various sources to constrain precipitation-based uncertainties (e.g., gauge-
corrected and/or Mountain Mapper MRMS), although the long processing time of these datasets
inhibits timely post-event assessments.
In addition to the above results focused primarily on the Dolan burn scar, a key feature of WRF-
Hydro is its ability to simulate the land surface hydrology of expansive geographic domains, e.g.,
NOAA runs the National Water Model over the entire continental U.S. Development of tools
capable of regional susceptibility assessments is crucial, particularly in a wildfire-prone region
like California, due to the large spatial scale, diverse morphology, and often tight spatial gradients
of precipitation events and their interactions with geographically widespread wildfire burn scars.
For example, landfalling ARs are often long (1000s of km) filament-like systems with
heterogeneous intensity gradients along their length. As a demonstration of wide geographic
applicability, we assess susceptibility over our full model domain which includes more than 10,000
catchments and a number of 2020 wildfire burn scars in addition to the Dolan burn scar (Fig 11).
The domain-wide analysis reveals elevated peak discharge, i.e., elevated susceptibility, in areas of
high precipitation and in burned terrains (Figs. 11a–c). We highlight channelized and catchment-
area normalized debris flow susceptibility in non-Dolan burn scar sites in Figs. 11d–g. In an
operational forecast context, the ability to simulate landslide and debris flow susceptibilities and
hazards over numerous catchments at meteorologically appropriate scales represents a step-change
in the field. We argue that our demonstration of WRF-Hydro's debris flow susceptibility hindcast

capabilities should motivate further exploration and development for potential use in operational hazard forecasting.

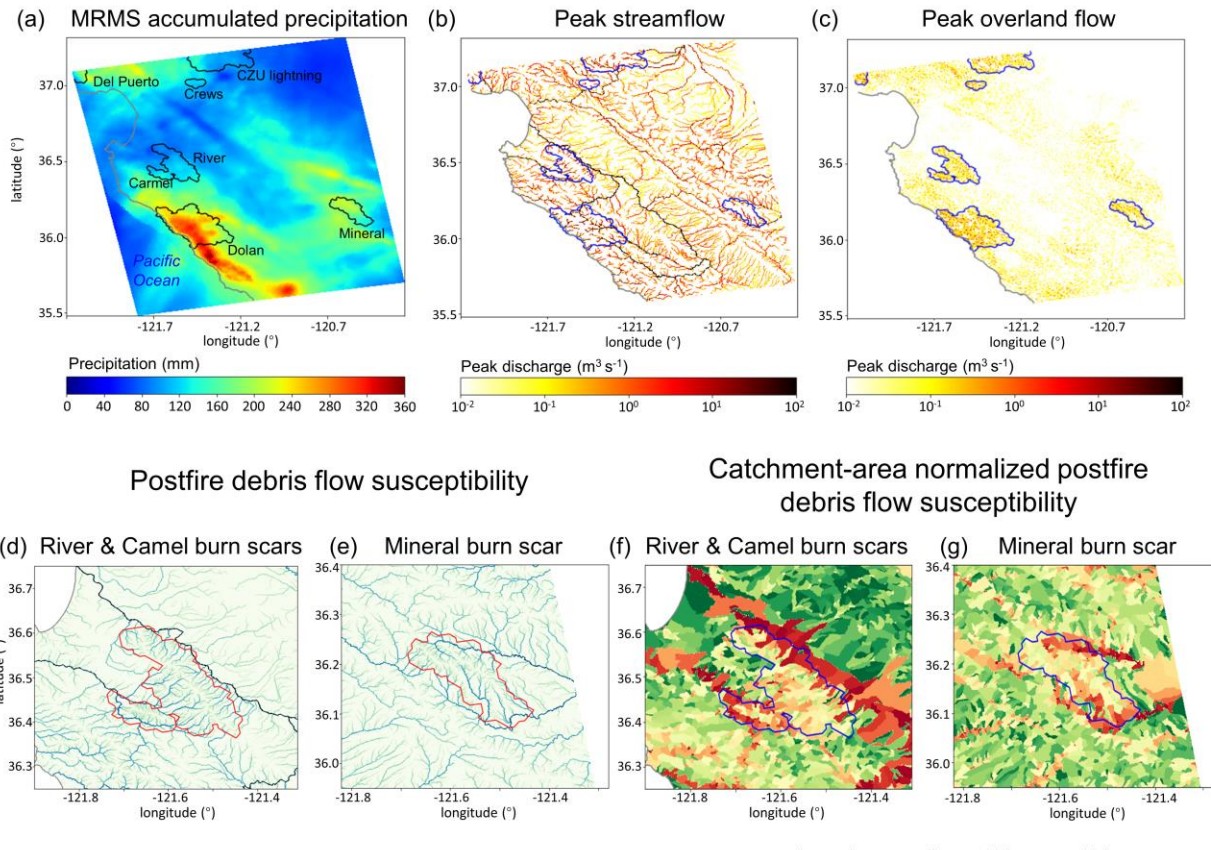

**Fig. 11|** MRMS accumulated precipitation and peak discharge informed regional debris flow susceptibility. (a) MRMS accumulated precipitation during January 27th 00:00 to 29th 23:00 over the model domain (mm; shading). Names of burn scars are labeled in black. (b) Peak streamflow ($m^3 s^{-1}$; yellow-to-red shading) and (c) peak overland flow from 27th 00:00 to 28th 12:00 over the model domain ($m^3 s^{-1}$; yellow-to-red shading). (d)–(e) Stream-level postfire debris flow susceptibility as Fig. 9b but for River and Camel burn scars. (f)–(g) Catchment-area normalized debris flow susceptibility as Fig. 9e but for River and Camel burn scars. Wildfire perimeters of

2020 wildfire season are outlined in black in (a), in blue in (b), (c), (f), and (g), and in red in (d)
and (e). The coastline of California is depicted in grey.
In addition to investigating the operationalization of WRF-Hydro's natural hazard prediction
capabilities, we note that with additional work our susceptibility-focused methodology could be
advanced to the level of hazard assessment, in line with current USGS debris flow products. The
USGS Emergency Assessment of Postfire Debris-flow Hazard predicts debris flow volume and
likelihood. To advance from susceptibility to hazard assessment, our methodology would need to
incorporate both debris flow volume estimates and occurrence likelihoods. In the following, we
highlight research directions that could help advance our susceptibility-focused methodological
framework. The first capability to develop would be a runoff-generated debris flow model that
couples hydrologic and sediment erosion and transport processes to help characterize postfire
debris flow volumes. Indeed, previous efforts have demonstrated the capacity to couple WRF-
Hydro with sediment flux models (Yin et al., 2020; Shen et al., 2021). In addition to sediments,
burn scar ash can comprise a substantial fraction of the total debris flow volume (e.g., Reneau et
al., 2007). As such, efforts to constrain ash availability and entrainment in hydrologic flows could
prove fortuitous in hazard assessment and prediction efforts. A second capability in need of
development is the use of WRF-Hydro to identify debris flow triggering time and location by
employing a domain-specific rainfall ID threshold trained with historic landslide inventory and
triggering rainfall events (Tognacca et al., 2000; Gregoretti & Dalla Fontana, 2007, 2008) or a
newly developed dimensionless discharge and Shields stress threshold (Tang et al., 2019a;
McGuire & Youberg, 2020). While in this study we do not attempt to simulate debris flow
dynamics such as triggering, we note that WRF-Hydro is capable of simulating overland flow and
streamflow at higher spatiotemporal resolutions [on scales that are similar to other debris flow
mechanistic studies such as Rengers et al. (2016), McGuire et al. (2016, 2017), and Tang et al.
(2019a, 2019b)]. Therefore, WRF-Hydro's capability to simulate the triggering processes of
runoff-generated debris flows is potentially only limited by the spatiotemporal resolution of
precipitation forcing and computing resources.
In addition to constraining postfire debris flow volumes and occurrence likelihoods, WRF-Hydro's
application in debris flow studies could be advanced via concerted engagement with uncertainties
that are both external (meteorological forcing data) and internal (physical parameters) to the model.
Previous studies have demonstrated that precipitation is often the largest source of uncertainty in
hydrologic predictive models (Hapuarachchi et al., 2011; Alfieri et al., 2012). Engagement with
precipitation forcing uncertainties in past, near-term, and future contexts could provide
probabilistic nuance to natural hazard investigations. For example, (a) debris flow hindcast studies
could use a diversity of precipitation datasets to isolate precipitation-derived debris flow
uncertainties in historic events, (b) operational forecast efforts could utilize ensemble-based
weather forecast data to inform likelihood statements in debris flow hazard assessments, and (c)
probabilistic projections of debris flow likelihood in future climates could assess and partition

uncertainties derived from emission pathway, model structure, or internal variability effects on meteorological forcings (Nikolopoulos et al., 2019; Hawkins & Sutton, 2009; Deser et al., 2020). Uncertainties internal to WRF-Hydro are also ripe for investigation. Probabilistic predictions crafted from an ensemble of perturbed model physics simulations have been used to predict rainfall-triggered shallow landslides (Raia et al., 2014; Canli et al., 2018; Zhang et al., 2018). Similar efforts using WRF-Hydro could target post-wildfire debris flows.

Lastly, the above discussion of potential WRF-Hydro applications and advancements speaks to the adaptability and customization of this open-source numerical model. An additional layer of WRF-Hydro's adaptability concerns its geographic focus. While we calibrate and use the model over a central California domain, the choice of geographic footprint is only limited by the availability of requisite initial and boundary conditions, environmental observations for calibration, and computational resources. For use in non-central California domains, we recommend calibration beginning with the default version of the model. Given the ecological and geological diversity of locations that experience wildfires and debris flows, it is likely that calibrations distinct from those reported here will be needed in different regions. For example, soil sealing effects, infiltration, and runoff in wetter and more vegetated locations, such as Oregon, USA, behave differently than those in central California (Palmer, 2022). As such, calibration of relevant model parameters (e.g., saturated hydraulic conductivities) should be based on a physics-informed approach that accounts for local environmental conditions and hydrologic behaviors. Indeed, given the ability to simulate large heterogeneous geographic domains, it is likely that different regions within a given domain may require different calibration schemes. As WRF-Hydro is fully distributed, spatially heterogeneous calibrations are non-problematic. This spatial adaptability may prove particularly helpful in post-wildfire debris flow hazard assessments when considering multiple generations of wildfires and variable degrees of burn scar severity and recovery.

**7 Conclusion**

Here we augment WRF-Hydro to assess regional postfire debris flow susceptibility. Our methodology involves output of simulated overland flow data and alteration of the model's representation of burn scars. In this application we have balanced the computational cost of a regional domain with our choice of resolved spatial resolution for terrain routing and overland flow calculations (100 m). However, WRF-Hydro has previously been applied to smaller domains at higher terrain routing resolutions (~30 m). Future work could assess the use of the model to study burn scar hydrology at finer spatial scales, should the application warrant and should underlying data at sufficient resolution exist. Other potential applications of our augmented model framework include alpine areas and steep hillslopes with sparse vegetation where runoff-generated debris flows dominate over landslide-initiated ones (Davies et al., 1992; Coe et al., 2003, 2008). Furthermore, our burn scar parameter changes are performed to Noah-MP, which is the core land surface component of the NCEP Global Forecast System (GFS) and Climate Forecast System

(CFS), thus the findings presented herein, are likely to prove useful in the broader worlds of
forecast meteorology and climate science. In addition, here WRF-Hydro is driven by historical
precipitation and meteorological data, i.e., in hindcast mode. However, this modeling framework
could also be employed to project hazards under future climatic conditions (e.g., Huang et al.,
2020a), or given its relatively low computational expense, in operational forecast mode. Indeed,
modern ensemble-based meteorological forecasting could provide high spatiotemporal forcing
data with which disaster preparedness managers could probabilistically assess debris flow hazard
potential, and issue advanced life and property saving warnings.

**Appendix A**
**Text A1. Multi-Radar/Multi-Sensor System (MRMS) radar-only precipitation estimate and**
**uncertainty**
MRMS is a precipitation product that covers the contiguous United States (CONUS) on 1-km grids.
It combines precipitation estimates from sensors and observational networks (Zhang et al.,
2011, 2014, 2016), and is produced at the National Centers for Environmental Prediction (NCEP)
and distributed to National Weather Service forecast offices and other agencies. Input datasets
used to produce MRMS include the U.S. Weather Surveillance Radar-1988 Doppler (WSR-88D)
network and Canadian radar network, Parameter-elevation Regressions on Independent Slopes
Model (PRISM; Daly et al., 1994, 2017), Hydrometeorological Automated Data System (HADS)
gauge data with quality control (Qi et al., 2016), and outputs from numerical weather prediction
models. There are four different MRMS quantitative precipitation estimates (QPE) products
incorporating different input data or combinations: radar only, gauge only, gauge-adjusted radar,
and Mountain Mapper. One caveat of using MRMS is that weather radars are problematic in
accurately capturing rainfall in high mountainous areas due to beam blocking by the orography
(Anagnostou et al., 2010; Germann et al., 2007), and gauge-corrected and Mountain Mapper
MRMS are superior and preferred. However, for our study period (i.e., January 1–31, 2021), the
gauge-corrected and Mountain Mapper MRMS are not available (as of May 2022).

We acknowledge that precipitation data has uncertainties. Use of different precipitation products
may produce different results. A study comparing different gridded precipitation datasets including
satellite-based precipitation data, gauge dataset, and multi-sensor products revealed large
uncertainties in precipitation intensity (Bytheway et al., 2020). However, comparing different
precipitation datasets to characterize uncertainties is beyond the scope of this study. MRMS
provides gridded precipitation at high temporal (hourly) and spatial (1-km) resolutions, making it
a useful tool to demonstrate the utility of WRF-Hydro in post-wildfire debris flow susceptibility
assessments.
**Appendix B**

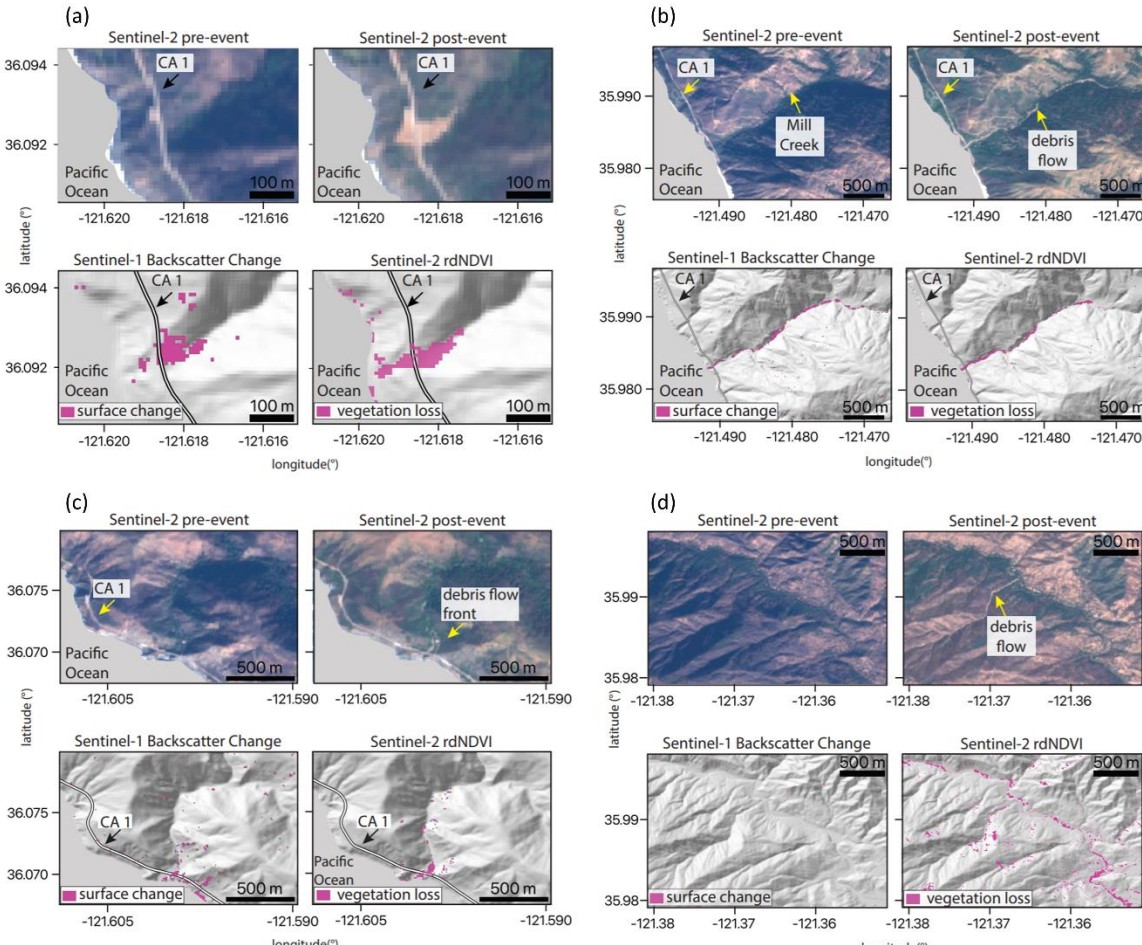

**Fig. B1** Optical- and SAR-based remote sensing data of four debris flows. Optical data from
Sentinel-2 show pre- and post-debris flow imagery in real color. rdNDVI calculated from the
Sentinel-2 data show a decrease in vegetation corresponding to debris flow locations. Sentinel-1
backscatter change shows the change in ground surface properties determined by calculating the
log ratio of pre- and post-event SAR images. The pre-event, post-event satellite images, Sentinel-
1 Backscatter, and Sentinel-2 rdNDVI change at (a) Rat Creek, (b) Mill Creek, (c) Big Creek, and
(d) Nacimiento.

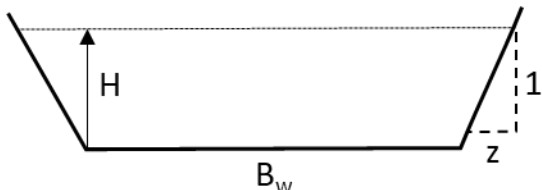

**Fig. B2** Schematic trapezoidal shape and related parameters of channels in WRF-Hydro. $B_w$ is
the channel bottom width (m), $z$ is the channel side slope (m), and $H$ is water elevation (m). The
cross-sectional area of flow is calculated as $(B_w + H\,z)H$.

**Table B1 Parameters of trapezoidal channels in WRF-Hydro.**

| Stream order | Channel bottom width $B_w$ (m) | Channel side slope $z$ (m) | Manning's roughness coefficient $n$ |
|:---:|:---:|:---:|:---:|
| 1 | 1.5 | 3 | 0.33 |
| 2 | 3 | 1 | 0.21 |
| 3 | 5 | 0.5 | 0.09 |
| 4 | 10 | 0.18 | 0.06 |
| 5 | 20 | 0.05 | 0.04 |
| 6 | 40 | 0.05 | 0.03 |
| 7 | 60 | 0.05 | 0.02 |
| 8 | 70 | 0.05 | 0.02 |
| 9 | 80 | 0.05 | 0.01 |
| 10 | 100 | 0.05 | 0.01 |


**Table B1** Parameters of the trapezoidal channels in WRF-Hydro including channel bottom width
$B_w$ (m), channel side slope $z$ (m), and Manning's roughness coefficient $n$.


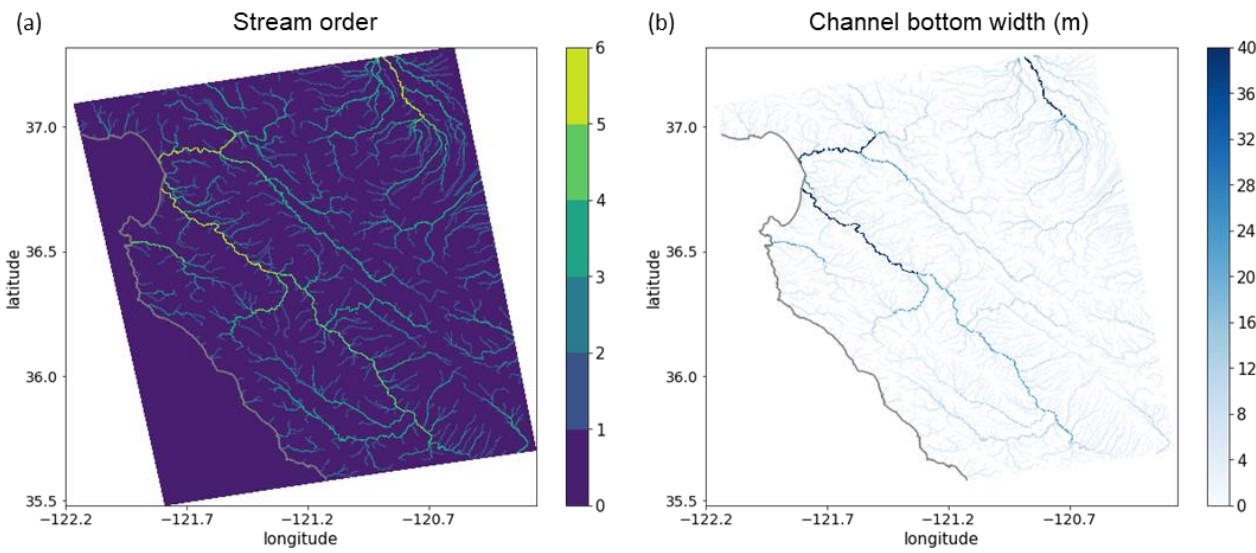


Fig. B3 (a) Stream order defined by the USGS 30-m DEM in our WRF-Hydro model domain
and (b) the channel bottom width (m) which is a function of stream order (Table B1).


 *Table B2*

 *MODIS IGBP 20-category land cover type and properties in Noah-MP LSM*

| Land cover code | Land cover type | Canopy height (m) | Max carboxylation rate at 25°C ($\mu mol\, CO_2/(m^2 \cdot s)$ ) | Overland flow roughness |
|---|---|---|---|---|
| 1 | Evergreen Needleleaf Forest | 20 | 50 | 0.2 |
| 2 | Evergreen Broadleaf Forest | 20 | 60 | 0.2 |
| 3 | Deciduous Needleleaf Forest | 18 | 60 | 0.2 |
| 4 | Deciduous Broadleaf Forest | 16 | 60 | 0.2 |
| 5 | Mixed Forests | 16 | 55 | 0.2 |
| 6 | Closed Shrublands | 1.1 | 40 | 0.055 |
| 7 | Open Shrublands | 1.1 | 40 | 0.055 |
| 8 | Woody Savannas | 13 | 40 | 0.055 |
| 9 | Savannas | 10 | 40 | 0.055 |
| 10 | Grasslands | 1 | 40 | 0.055 |
| 11 | Permanent wetlands | 5 | 50 | 0.07 |
| 12 | Croplands | 2 | 80 | 0.035 |
| 13 | Urban and Built-Up | 15 | 0 | 0.025 |
| 14 | Cropland/natural vegetation mosaic | 1.5 | 60 | 0.035 |
| 15 | Snow and Ice | 0 | 0 | 0.01 |
| 16 | Barren or Sparsely Vegetated | 0 | 0 | 0.035 |
| 17 | Water | 0 | 0 | 0.005 |
| 18 | Wooded Tundra | 4 | 50 | 0.055 |
| 19 | Mixed Tundra | 2 | 50 | 0.055 |
| 20 | Barren Tundra | 0.5 | 50 | 0.055 |


**Table B2** MODIS IGBP 20-category land cover type and properties in Noah-MP LSM.


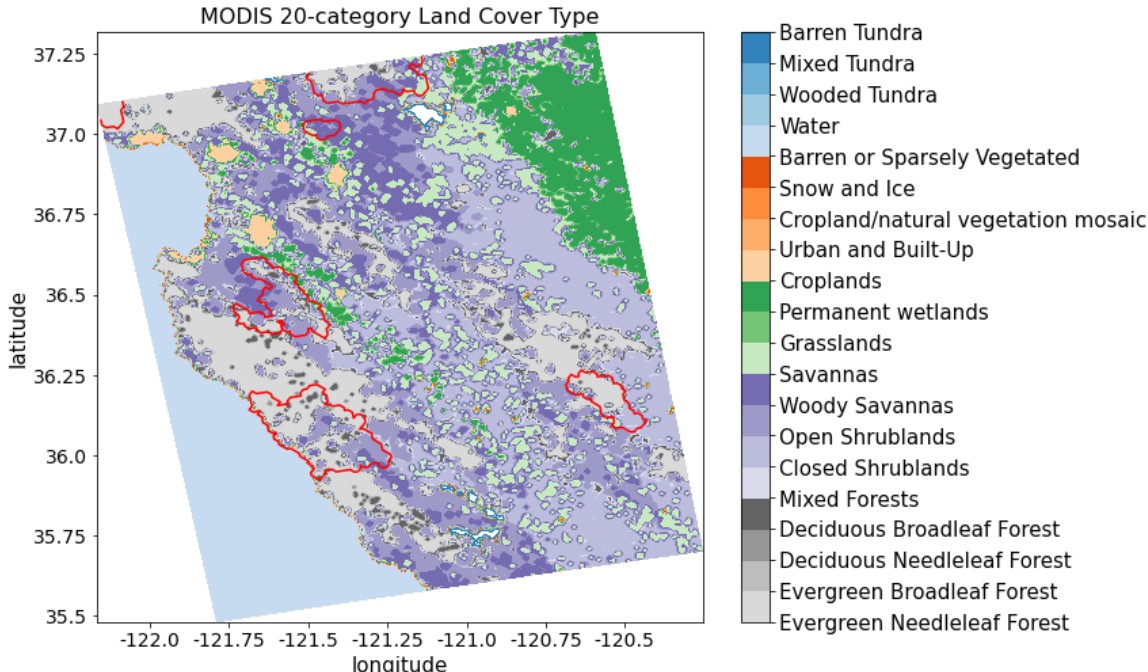

**Fig. B4** MODIS IGBP 20-category land cover type in the model domain. Red polylines are 2020 wildfire burn scar perimeters.

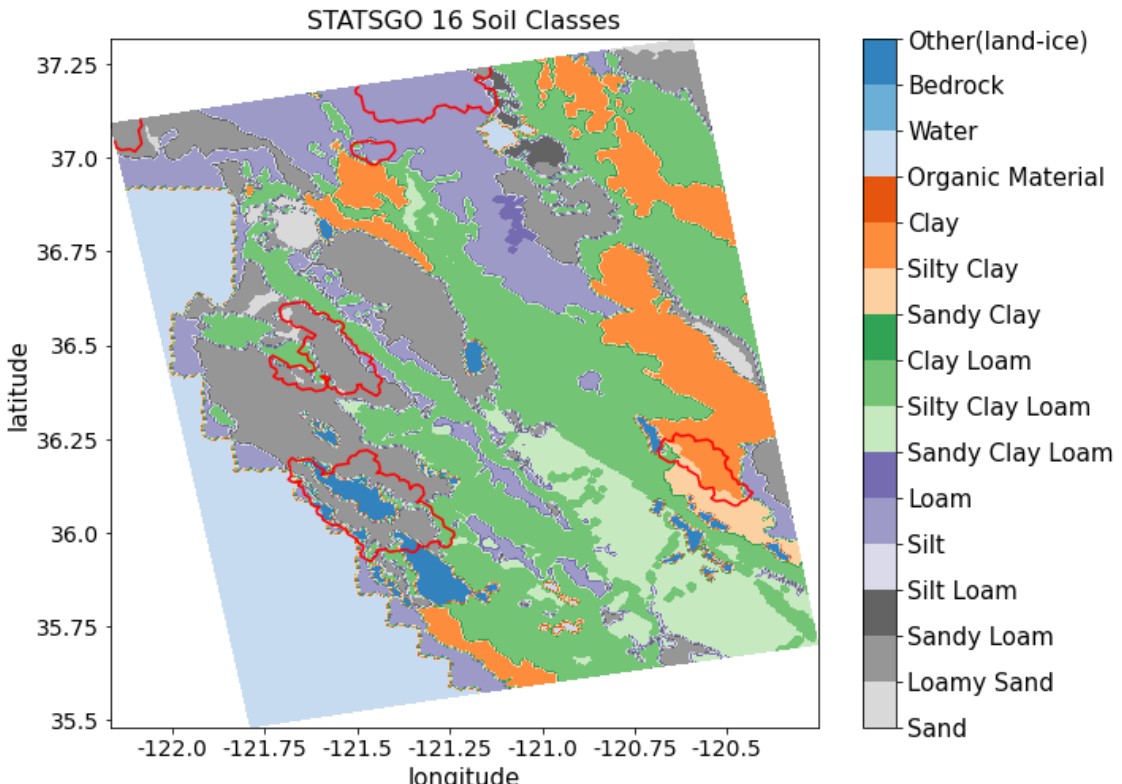

Fig. B5 1-km STATSGO data with 16 soil texture types. Red polylines are 2020 wildfire burn scar perimeters.

*Table B3*
*Default and calibrated soil parameters in WRF-Hydro*

| Soil type | Default | | | After calibration | | |
|---|---|---|---|---|---|---|
| | Grain size distribution index | Porosity | Saturated hydraulic conductivity $(m\ s^{-1})$ | Grain size distribution index | Porosity | Saturated hydraulic conductivity $(m\ s^{-1})$ |
| Sand | 2.79 | 0.339 | 4.66E-5 | 2.51 | 0.315 | |
| Loamy sand | 4.26 | 0.421 | 1.41E-5 | 3.83 | 0.392 | |
| Sandy loam | 4.74 | 0.434 | 5.23E-6 | 4.27 | 0.404 | |
| Silt loam | 5.33 | 0.476 | 2.81E-6 | 4.80 | 0.442 | |
| Silt | 3.86 | 0.484 | 2.18E-6 | 3.47 | 0.450 | |
| Loam | 5.25 | 0.439 | 3.38E-6 | 4.73 | 0.408 | |
| Sandy clay loam | 6.77 | 0.404 | 4.45E-6 | 6.09 | 0.376 | |
| Silty clay loam | 8.72 | 0.464 | 2.03E-6 | 7.85 | 0.432 | $1.5 \times 10^{-7}\ m\ s^{-1}$ for all the burn scars, and original values elsewhere. |
| Clay loam | 8.17 | 0.465 | 2.45E-6 | 7.35 | 0.432 | |
| Sandy clay | 10.73 | 0.406 | 7.22E-6 | 9.66 | 0.378 | |
| Silty clay | 10.39 | 0.468 | 1.34E-6 | 9.35 | 0.435 | |
| Clay | 11.55 | 0.468 | 9.74E-7 | 10.40 | 0.435 | |
| Organic material | 5.25 | 0.439 | 3.38E-6 | 4.73 | 0.408 | |
| Water | 0.00 | 1.00 | 0.00 | 0.00 | 1.00 | |
| Bedrock | 2.79 | 0.200 | 1.41E-4 | 2.51 | 0.186 | |
| Other | 4.26 | 0.421 | 1.41E-5 | 3.83 | 0.392 | |
| Playa | 11.55 | 0.468 | 9.74E-7 | 10.40 | 0.435 | |
| Lava | 2.79 | 0.200 | 1.41E-4 | 2.51 | 0.186 | |
| White sand | 2.79 | 0.339 | 4.66E-5 | 2.51 | 0.315 | |


**Table B3** Soil parameters in default and calibrated WRF-Hydro. Default soil parameters in WRF-
Hydro are adapted from the soil analysis by Cosby et al. (1984). Grain size distribution index and
soil porosity are altered from default values during the global soil moisture calibration. Saturated
hydraulic conductivity is altered from default values during the streamflow calibration.


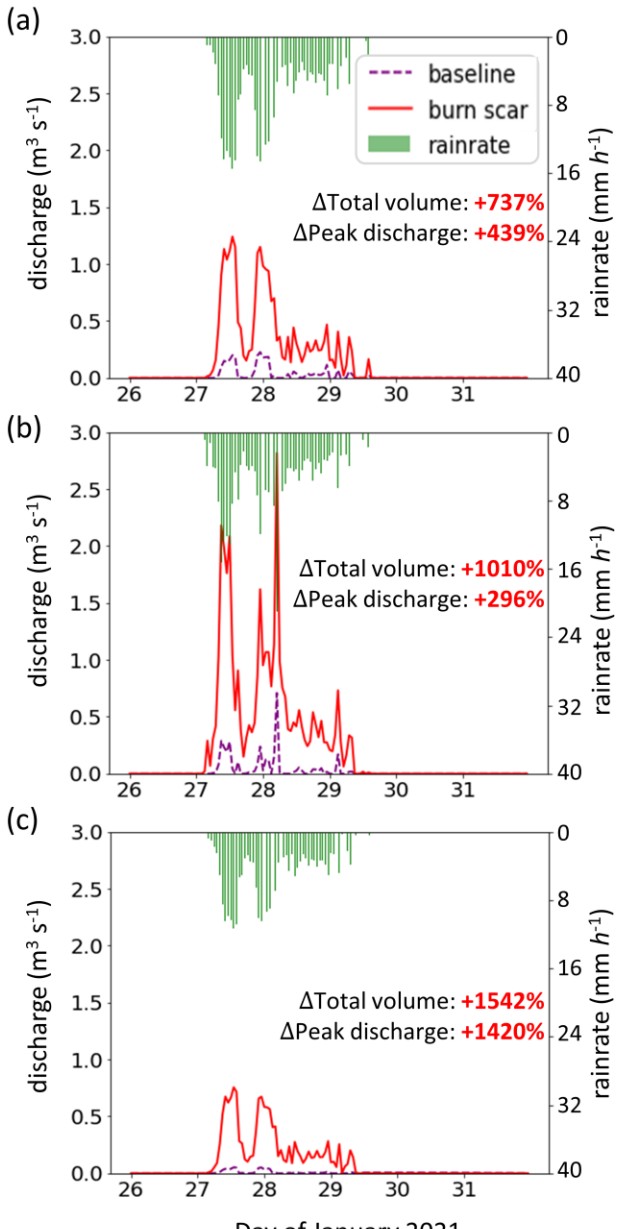

**Fig. B6** WRF-Hydro simulated discharge time-series at four debris flow source areas. (a)–(c)
MRMS precipitation (green bars) and simulated discharge time-series for January 26[th] 00:00 to
31[st] 23:00 at Mill Creek, Big Creek, and Nacimiento debris flow source areas (black circles in Fig.
7b–d) in baseline (purple dashed line) and burn scar simulation (red line).

*Table B4*
*The total runoff volume, peak discharge, and peak timing at debris-flow source areas*

| Site name | Baseline simulation | | | Burn scar simulation | | |
|---|---|---|---|---|---|---|
| | Total volume (m³) | Peak discharge (m³ s⁻¹) | Peak timing | Total volume (m³) | Peak discharge (m³ s⁻¹) | Peak timing |
| **Mill Creek** | 10,023 | 0.23 | 27th 23:00 | 83,853 (+737%) | 1.24 (+439%) | 27th 13:00 |
| **Big Creek** | 11,611 | 0.71 | 28th 05:00 | 128,879 (+1010%) | 2.81 (+296%) | 28th 05:00 |
| **Nacimiento** | 3,031 | 0.05 | 27th 13:00 | 49,792 (+1542%) | 0.76 (+1420%) | 27th 13:00 |


**Table B4** The total runoff volume, peak discharge, and peak timing in the baseline and burn scar
simulations from January 27th 00:00 to 31st 23:00 at source areas of Rat Creek, Mill Creek, Big
Creek, and Nacimiento debris flows (black circles in Fig. 7b–d). The percent change of the total
volume and peak discharge in the burn scar simulation relative to the baseline simulation are shown
in parentheses.


### Stream channel accumulated discharge volume

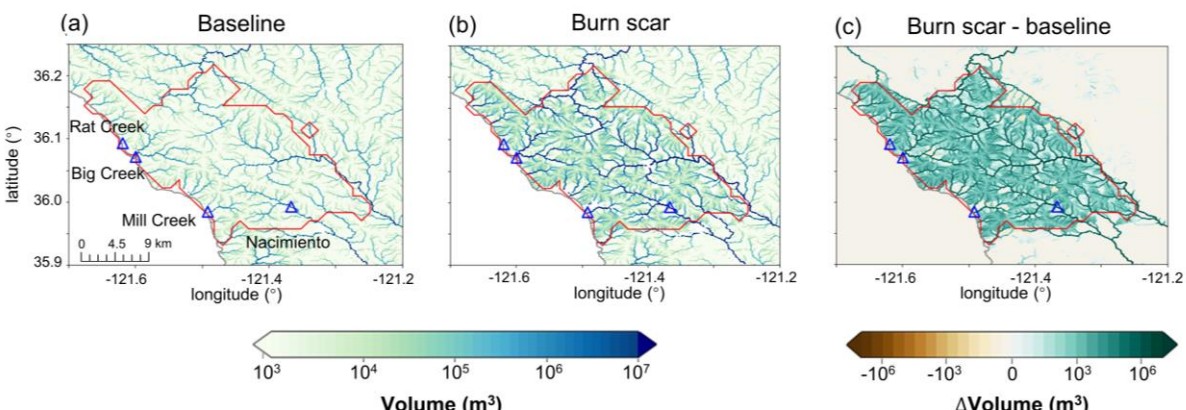

### Catchment-area normalized accumulated discharge volume

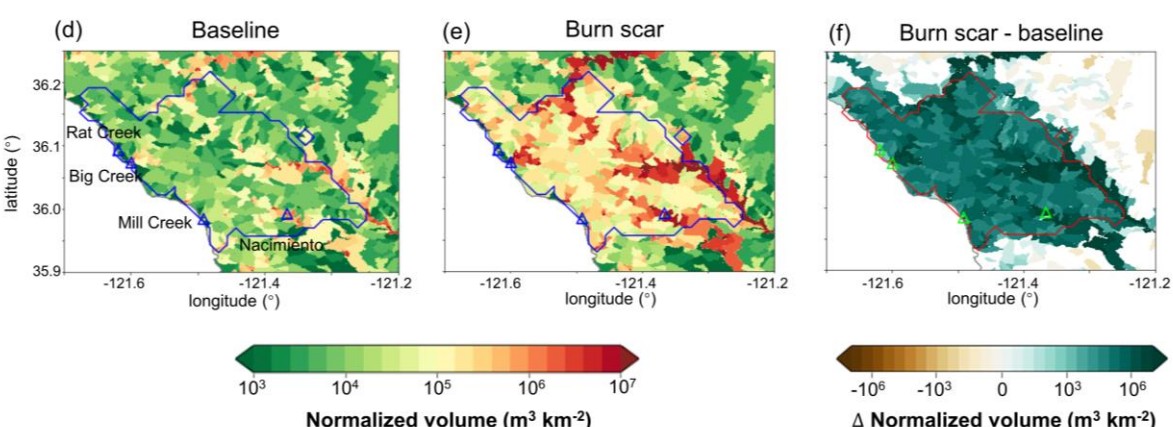


**Fig. B7** Accumulated discharge volume at individual stream level for the (a) baseline, (b) burn
scar, and (c) difference between burn scar and baseline simulations (m$^3$). Total discharge volume
is accumulated from January 27th 00:00 to 28th 12:00. (d)–(f) Normalized discharge volume by
catchment area at catchment level (m$^3$ km$^{-2}$; shading; Santi & Morandi, 2013). For each catchment,
the discharge volume is accumulated at the catchment outlet from January 27th 00:00 to 28th 12:00
divided by catchment area. Triangles stand for debris flow deposition locations and are annotated
in (a) and (d).






*Table B5*
*Statistics of catchment area-normalized discharge volume in baseline and burn scar simulations*

| | mean | std | 5P | 25P | 50P | 75P | 95P |
|---|---|---|---|---|---|---|---|
| **Baseline simulation** ($m^3$ $km^{-2}$) | 380k | ±1.6M | 0.6k | 3.7k | 13k | 120k | 2.1M |
| **Burn scar simulation** ($m^3$ $km^{-2}$) | 5.5M | ±23.0M | 1.5k | 30.7k | 135k | 1.3M | 29.1M |
| **Relative percent change** | 1300% | 1400% | 148% | 725% | 924% | 980% | 1300% |

**Table B5** Statistics, including the mean, standard deviation (std), 5P, 25P, 50P, 75P, and 95P, of
the catchment-area normalized discharge volume for all the 404 basins within the Dolan burn scar
in the baseline and burn scar simulation and their relative percent changes.








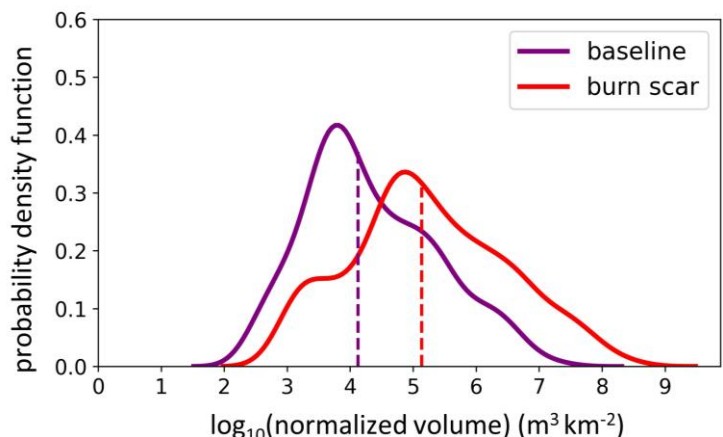

**Fig. B8** Distributions of accumulated discharge volumes at the outlet of the 404 catchments normalized by upstream catchment areas within Dolan burn scar in the baseline simulation (purple line) and in the burn scar simulation (red line). Dashed vertical lines indicate median values.

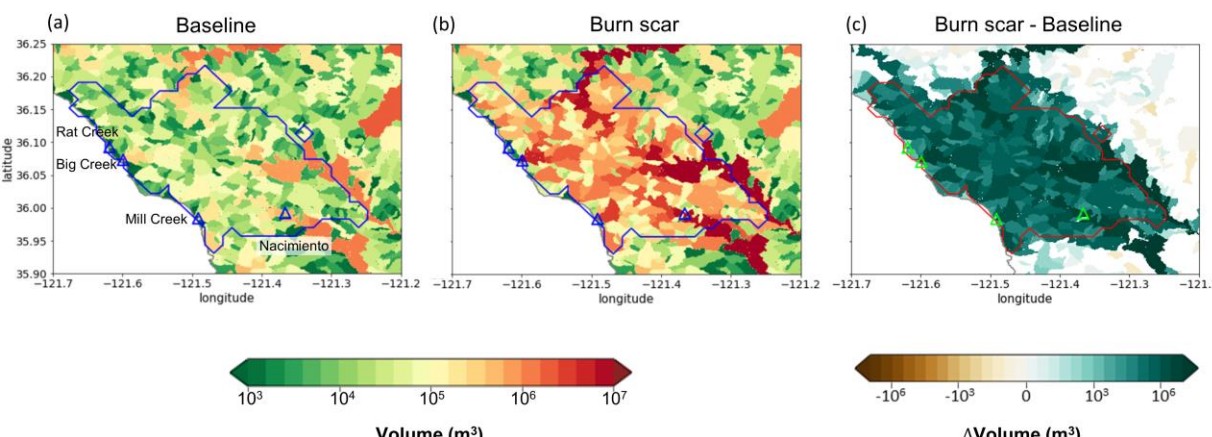

979

980

**Fig. B9** Non-normalized accumulated discharge volume at catchment level in the (a) baseline simulation, (b) burn scar simulation, and (c) the difference between the burn scar and baseline simulations (m$^3$; shading). For each catchment, the discharge volume is accumulated at the catchment outlet from January 27$^{th}$ 00:00 to 28$^{th}$ 12:00. Triangles stand for debris flow deposition locations and are annotated in (a).


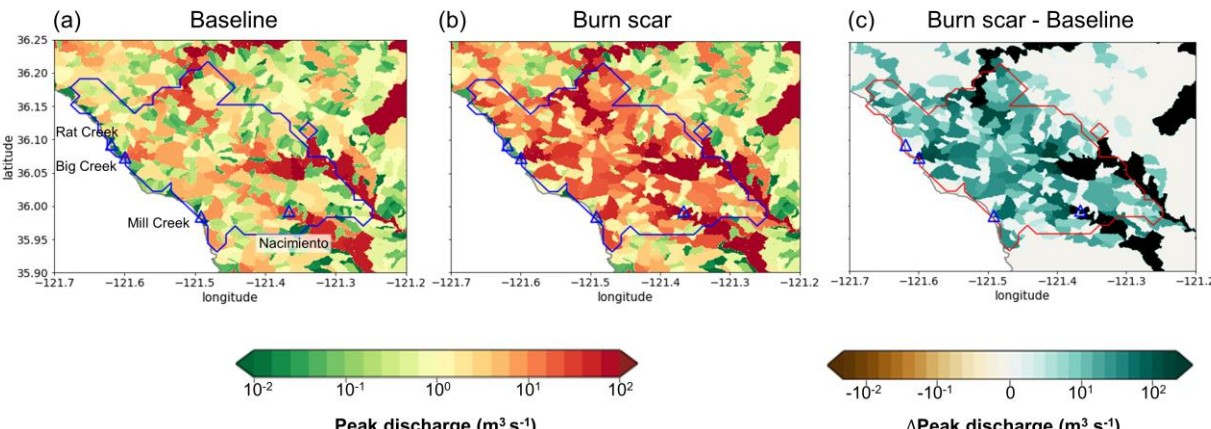


**Fig. B10** Non-normalized peak discharge at catchment level in the (a) baseline simulation, (b) burn scar simulation, and (c) the difference between the burn scar and baseline simulations ($m^3 s^{-1}$; shading). For each catchment, the peak discharge is the maximum discharge rate at the catchment outlet from January 27th 00:00 to 28th 12:00. Triangles stand for debris flow deposition locations and are annotated in (a).






**Data availability statement**

The NLDAS-2 reanalysis forcing data is publicly available at NASA GES DISC: https://disc.gsfc.nasa.gov/datasets?keywords=NLDAS. A detailed description can be found at https://ldas.gsfc.nasa.gov/nldas/v2/forcing. The MRMS radar-only precipitation estimate is publicly available at: https://mtarchive.geol.iastate.edu/. A description can be found at https://www.nssl.noaa.gov/projects/mrms/. The PSL in-situ soil moisture data is publicly available at: https://psl.noaa.gov/data/obs/datadisplay/. The USGS streamflow is publicly available at: https://waterdata.usgs.gov/nwis/. The wildfire perimeter shapefiles are downloadable at: https://data-nifc.opendata.arcgis.com/search?collection=Dataset. The remote sensing data used in this manuscript were provided by the European Space Agency (ESA) Copernicus program and accessed on Google Earth Engine (https://code.earthengine.google.com). All processed data required to reproduce the results of this study are archived on Zenodo at http://doi.org/10.5281/zenodo.5544083.

## Code availability statement

The modified WRF-Hydro Fortran code and instructions to output the overland flow at terrain routing grid can be downloaded at https://github.com/NU-CCRG/Modified-WRF-Hydro.

HazMapper v1.0 is available at https://hazmapper.org/. The SAR backscatter change method code is available at https://github.com/MongHanHuang/GEE_SAR_landslide_detection.

## Author contribution

Conceptualization: CL, ALH, & DEH; Simulation and model analysis: CL; JW & WY model methodological development. Remote sensing analysis: ALH; Field Observations: NJF; GIS assistance: YX; Funding acquisition: GB & DH; CL wrote the original draft and all authors reviewed and edited the manuscript.

## Competing interests

The authors declare that they have no conflict of interest.

## Acknowledgments

C.L., A.L.H., J.W., X.L., G.B., and D.E.H. acknowledge support from NSF PREEVENTS #1848683. We acknowledge high-performance computing support from Cheyenne (doi:10.5065/D6RX99HX) provided by NCAR's Computational and Information Systems Laboratory, sponsored by the National Science Foundation. We thank P. Santi, an anonymous reviewer, and the editor for formal reviews, and F. K. Rengers for informal comments.

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
