# Peer review of "Augmentation of WRF-Hydro to Simulate Overland Flow- and Streamflow-Generated Debris Flow Susceptibility in Burn Scars"

_Natural Hazards and Earth System Sciences, 2021_

## Referee Comment (RC2)

[referee-annotated manuscript omitted]

---

## Author Comment (AC2)

General Comments

The paper is very well written and easy to follow, and it is a nice integration of modern modeling techniques and data for use in debris flow analysis.

Response: We thank Dr. Paul Santi for his constructive comments of our manuscript. We plan to make specific changes to the manuscript in response to each of his comments, and believe the manuscript will be significantly improved as a result.

I think a slight change in the declared focus of the paper will better highlight its value. Allow me to explain. At many points in the paper, the authors have gone to a lot of trouble to set up, run, and calibrate models that basically demonstrate the same things that have been said (and quantified) in other papers using much simpler analyses: debris flow volume and discharge increase multifold in burned areas, the hazard is concentrated in stream channels, and there is a lag between rainfall peaks and flow events, for example. However, the authors' analyses provide some information that has not been clearly shown before. Importantly, they are able to create calibrated time graphs of streamflow and discharge. Also, they are able to compare their models with the USGS post-wildfire assessments to show differences (they refer to this in lines 652-656, but don't give details of the analysis). I think the paper would be stronger if they acknowledge early on that other research has demonstrated (and quantified) changes in volume, discharge, and lag. Then they could focus on the advantages offered by a more sophisticated, calibrated model.

Response: We thank Dr. Santi for his suggested re-framing of the manuscript. We agree that the current iteration of the manuscript is heavily focused on the Dolan wildfire burn scar case study, and that the manuscript would be better served if it had a greater focus on our modeling advance and its potential for future applications and advancements. In our revision, we will re-focus the latter half of our abstract to better emphasize the advance and potential of using WRF-Hydro in debris flow studies and forecasts. As requested, we will also acknowledge (and cite) early in the manuscript that others have demonstrated similar debris flow hydrologic behaviors using numerical models in limited domains, while emphasizing the value added by our regionalized and fully-distributed hydrologic simulations. In addition, rather than focusing on our case study results at the start of the Discussion section, we will provide a more general summary emphasizing the advances provided by using a model like WRF-Hydro to investigate debris flows.

I think the discussion should also include a section on applying the model elsewhere. Is it realistic to do this for other sites, or is it too dependent on specific calibration parameters? How could a practitioner do this type of analysis? What does it offer a scientist that they do not already know?

Response: This is another excellent suggestion, and has parallels with a recent news piece found in *Nature* (Palmer, 2022). WRF-Hydro can indeed be applied elsewhere. The model has been used to study a diverse range of hydrological processes in domains of varying size across the world. Use of WRF-Hydro and choice of spatial resolution is dependent on the existence of requisite boundary conditions, forcing files, and observational constraints. For studies focused on burn scar-debris flow dynamics, the model is again readily adaptable. In our approach, we demonstrate that the burn scar characteristics of a land surface can be set in the land surface model (i.e., reduced canopy height, overland roughness, carboxylation rate, and infiltration rate). Since parameter values will vary based on a myriad of factors (i.e., geography, climate, biome, soil properties, etc.), a major advantage of WRF-Hydro is the ability to modify and calibrate the underlying physical parameters as appropriate for each location. In our revised version of the manuscript, we will discuss the application of WRF-Hydro in other regions, and provide a roadmap for global community usage.

The discussion could also compare their model to the USGS model, using a modification to Figure 9, for example, to demonstrate and explain the important differences.

Response: Referee #2 has mentioned a key point that differentiates our susceptibility assessment from the USGS' hazard assessment. That is, the USGS statistical models are able to predict both probability and magnitude of debris flows, which makes them "hazard" assessments, whereas our model is focused on predicting which areas are subject to higher likelihood and should be referred to as "susceptibility" assessment. In the revision, we plan to explain this terminology and alter its usage throughout the manuscript and in the title. We will also discuss methods that could be employed to create probabilistic hazard assessments using WRF-Hydro, which would facilitate an apples-to-apples comparison with the USGS product.

Specific Comments

Section 5.4 - I don't feel that this is a strong section. It concludes that the hazards are greater in the burned area, and mostly in the channels, and that streamflow is elevated downstream in burned areas, which are not unique findings. Likewise, Figure 11 doesn't come across as strongly as previous figures. I suggest dropping this section.

Response: We somewhat agree with this suggestion. In our revision we will remove the regional discussion and Figure 11 from the results portion of the manuscript. However, one of the values added by WRF-Hydro is regional prediction and projection, which differs from more traditional single catchment simulations (e.g., McGuire et al., 2016, 2017). To highlight these capabilities, particularly from a future usage perspective, we will move Figure 11 to the discussion section and discuss why regional applications, particularly in an operational setting, provide value.

line 489 ff - an interesting note, your modeled discharge increases by 3 or 4 fold matches field measured changes published in Brunkal and Santi for large drainage basins (I could not find the area for your drainage basins, since you include normalized values, but I assume they are more than 5 km^2) (Brunkal, H. and Santi, P., 2017, "Consideration of the Validity of Debris-Flow Bulking Factors," Environmental and Engineering Geoscience, DOI: 10.2113/EEG-1774). See Figure 3 of this paper.

Response: In the revision, we will cite this paper in the discussion section to highlight the similarity.

Technical Corrections

Figures 1, 7. and 9 could benefit from a bar scale.

Response: In the revision we will add scale bars to Figures 1, 7, and 9.

Figure 9 - the legend is hard to understand. I assume the first bar is volume and the second is normalized volume?

Response: Yes, the first bar is volume and the second is normalized volume. We will revise the legend to make this clearer.

We thank Dr. Paul Santi again for his careful review and constructive comments.

**References**

McGuire, L. A., J. W. Kean, D. M. Staley, F. K. Rengers, and T. A. Wasklewicz (2016), Constraining the relative importance of raindrop- and flow-driven sediment transport mechanisms in postwildfire environments and implications for recovery time scales, *J. Geophys. Res. Earth Surf.*, 121, 2211–2237, doi:10.1002/2016JF003867

McGuire, L. A., Rengers, F. K., Kean, J. W., and Staley, D. M. (2017), Debris flow initiation by runoff in a recently burned basin: Is grain-by-grain sediment bulking or en masse failure to blame?, *Geophys. Res. Lett.*, 44, 7310–7319, doi:10.1002/2017GL074243.

Palmer, J. (2022). The devastating mudslides that follow forest fires. *Nature.* https://www.nature.com/articles/d41586-022-00028-3.

---

## Author Comment (AC3)

The manuscript deals with assessment of the debris flow hazard in burned areas through simulations that used high-resolution weather radar-derived precipitation. The manuscript has several interesting points, and overall is well written. It is certainly worth to be considered for publication, but I have a couple of points which need to be clarified.

Response: We thank referee #2 for providing valuable comments for the improvement of the manuscript. We plan to make specific changes in response to each of these comments, and believe that the manuscript will be significantly improved as a result of these changes.

The first (and main) one regards the terminology used. I am afraid that, throughout the article, the term hazard is not used correctly. In my opinion, Authors are rather talking about susceptibility, and not hazard, the difference being that hazard should depict the probability of occurrence of a certain phenomenon not only spatially but also temporally. This latter issue (time) is not considered in the study. I suggest go back to the original definition by Varnes (1984) and UNESCO, and in later works as well, to clarify the meaning of susceptibility and hazard, and to change accordingly the terms in the manuscript.

Response: We thank the referee for highlighting the issue of our loose use of terminology, and we agree with the referee's assessment that our manuscript and methodology are primarily focused on susceptibility, rather than probabilistic hazard assessment. In our revision, we will adhere to the definition of terms summarized in Section 2.1 of Reichenback et al (2018), *A review of statistically-based landslide susceptibility models*, Earth-Science Reviews. Accordingly, we plan to change the word "hazard" to "susceptibility" throughout the manuscript and in the title. In addition, we will add a paragraph to the discussion section which discusses methods by which WRF-Hydro could move beyond "susceptibility" assessments to probabilistic "hazard" quantification. Methods for probabilistic advancement include systemic investigation of parameter uncertainties, use of ensemble-based precipitation data, and quantification of precipitation return intervals with intensity-duration-frequency (IDF) curves.

Another point which needs more details is the description of the debris flows. Authors talk about several debris flows that occurred, and start to cite them in section 2.1. However, a clear description of the events, in terms of geology, morphology, morphometry, volumes is never properly given. This should be done the first time debris flows are mentioned (possibly in section 2.1) to let the reader understand the main characters of the events. For instance, were these debris flows individual phenomena, or did they start from multiple source areas? Further, were they channelized or openslope? More geomorphological info would be useful to understand the conditions under which the debris flows initiated and developed. Only at page 18 some info are provided, but these should appear much before than that, and be well organized, rather than distributed in different parts of the manuscript.

Response: We sympathize with the referee's desire for more data on the debris flows highlighted in our manuscript. However, at present, there have been no systematic studies of these debris flows, so while we are quite confident that debris flows occurred at these locations based on field observations of the deposits and our remote sensing analyses, information about source areas (which are in extremely inaccessible locations) and volumes are not well constrained. We do know that David Cavagnaro et al. (https://agu.confex.com/agu/fm21/meetingapp.cgi/Paper/921613) have undertaken a huge effort to map debris flows in the Dolan Fire burn scar. I suspect this forthcoming work will be able to answer some of the referee's concerns in greater detail.

In our revised manuscript we plan to move the descriptions on the debris flows in section 5.2 to section 2.1, and use Google Earth to estimate the number of source regions and morphology to the best of our ability.

In addition, we will provide more information on the geological setting of debris flows using the USGS geologic map (https://mrdata.usgs.gov/geology/state/state.php?state=CA).

Other issues:

Figure 1 definitely needs a location map, showing where we are in California, and in USA. Authors give for granted that anybody knows the site, but for an international journal a location map is always necessary.

Response: In our revision we will add a location map of the USA which depicts the locations of California and the burn scar/debris flow region.

Throughout the manuscript, references should be listed in chronological order when more than two references are cited. Some incomplete or wrong references are present in the list. Please check at this regard the attached file. Eventually, some minor issues are indicated in the accompanying file.

Response: In our revision we will reorder the in-text citations chronologically and correct the references. We thank referee #2 for their very careful examination of references. We have checked the attached file and will address those minor issues accordingly.

Overall, I evaluate positively the manuscript, which however needs to clarify the points outlined above, and recommend minor revisions.

We thank referee #2 again for their careful review and positive comments.

**References:**

Cavagnaro D et al. (2021) Variability in hydrologic response to rainfall across a burn scar: observations from the Dolan Fire, California. AGU abstract. https://agu.confex.com/agu/fm21/meetingapp.cgi/Paper/921613

Reichenbach P, Rossi M, Malamud BD, Mihir M, Guzzetti F (2018) A review of statistically-based landslide susceptibility models. Earth-Science Reviews. 180:60-91 doi:https://doi.org/10.1016/j.earscirev.2018.03.001

---

## Author Response (AR1)

\*Response is in blue & Sentences or paragraphs after revision are in red

General Comments

The paper is very well written and easy to follow, and it is a nice integration of modern modeling techniques and data for use in debris flow analysis.

Response: We thank Dr. Paul Santi for his constructive comments of our manuscript. We have made specific changes to the manuscript in response to each of his comments, and believe the manuscript has been significantly improved as a result.

I think a slight change in the declared focus of the paper will better highlight its value. Allow me to explain. At many points in the paper, the authors have gone to a lot of trouble to set up, run, and calibrate models that basically demonstrate the same things that have been said (and quantified) in other papers using much simpler analyses: debris flow volume and discharge increase multifold in burned areas, the hazard is concentrated in stream channels, and there is a lag between rainfall peaks and flow events, for example. However, the authors' analyses provide some information that has not been clearly shown before. Importantly, they are able to create calibrated time graphs of streamflow and discharge. Also, they are able to compare their models with the USGS post-wildfire assessments to show differences (they refer to this in lines 652-656, but don't give details of the analysis). I think the paper would be stronger if they acknowledge early on that other research has demonstrated (and quantified) changes in volume, discharge, and lag. Then they could focus on the advantages offered by a more sophisticated, calibrated model.

Response: We thank Dr. Santi for his suggested re-framing of the manuscript. We agree that the previous iteration of the manuscript was heavily focused on the Dolan wildfire burn scar case study, and that the manuscript is better served now because we put a greater focus on our modeling advance and its potential for future applications and advancements according to Dr. Santi's suggestion. In our revision, we re-focused the latter half of our abstract to better emphasize the advance and potential of using WRF-Hydro in debris flow studies and forecasts.

The abstract now reads:

"In steep wildfire-burned terrains, intense rainfall can produce large volumes of runoff that can trigger highly destructive debris flows. However, the ability to accurately characterize and forecast debris-flow susceptibility in burned terrains using physics-based tools remains limited. Here, we augment the Weather Research and Forecasting Hydrological modeling system (WRF-Hydro) to simulate both overland and channelized flows and assess postfire debris flow susceptibility over a regional domain. We perform hindcast simulations using high-resolution weather radar-derived precipitation and reanalysis data to drive non-burned baseline and burn scar sensitivity experiments. Our simulations focus on January 2021 when an atmospheric river triggered numerous debris flows within a wildfire burn scar in Big Sur – one of which destroyed California's famous Highway 1. Compared to the baseline, our burn scar simulation yields dramatic increases in total and peak discharge, and shorter lags between rainfall onset and peak discharge, consistent with streamflow observations at nearby U.S. Geological Survey (USGS) streamflow gage sites. For the 404 catchments located in the simulated burn scar area, median catchment-area normalized discharge volume increases nine-fold compared to the baseline. Catchments with anomalously high catchment-area normalized discharge volumes correspond well with post-event field-based and remotely-sensed debris flow observations. We suggest that our regional post-fire debris flow susceptibility analysis demonstrates WRF-Hydro as a compelling new physics-based tool whose utility could be further extended via coupling to sediment erosion and transport models and/or ensemble-based operational weather forecasts.

Given the high-fidelity performance of our augmented version of WRF-Hydro, as well as its potential usage in probabilistic hazard forecasts, we argue for its continued development and application in post-fire hydrologic and natural hazard assessments."

As requested, we also acknowledged (and cited) in the manuscript that others have demonstrated similar debris flow hydrologic behaviors using numerical models in limited domains, while emphasizing the value added by our regionalized and fully-distributed hydrologic simulations. Rather than focusing on our case study results at the start of the discussion section, we provided a more general summary emphasizing the advances provided by using a model like WRF-Hydro to investigate debris flows.

The first paragraph of our revised discussion section in lines 639 – 656 now reads:

"Given the historic and growing frequency of wildfires in the western U.S. (Williams et al., 2019; Goss et al., 2020; Swain 2021) and globally (Flannigan et al., 2013; Jolly et al., 2015), developing tools to investigate, better understand, and potentially predict changes in burn scar hydrology and natural hazards at regional scales is critical. Here, we demonstrate the first use of WRF-Hydro to simulate the susceptibility of a burn scar to postfire debris flows during a landfalling AR. We augmented the default version of WRF-Hydro to output overland flow and to replicate burn scar behavior by adjusting vegetation type and infiltration rate parameters. WRF-Hydro simulations were validated against PSL soil moisture and USGS streamflow observations before we used simulated streamflow and overland flow volumes to characterize debris flow susceptibility. A comparison between baseline and burn scar simulations demonstrated that changes in hydraulic properties of burned areas causes drastic changes in surface flows, including faster discharge response times, and greater peak discharge and total volumes, consistent with findings from previous postfire hydrology studies (Anderson et al., 1976; Scott, 1993; Meixner & Wohlgemuth, 2003; Kean et al., 2011; Kinoshita & Hogue, 2015; Brunkal & Santi, 2016; Williams et al., 2022). At the catchment level, for the 404 catchments located within the Dolan burn scar, median catchment area-normalized volume increases nine-fold relative to the baseline. In addition, Mill Creek, Big Creek, and Nacimiento basins were simulated to have high-to-very high debris flow susceptibility, corresponding well with identified debris flow occurrences."

I think the discussion should also include a section on applying the model elsewhere. Is it realistic to do this for other sites, or is it too dependent on specific calibration parameters? How could a practitioner do this type of analysis? What does it offer a scientist that they do not already know?

Response: This is another excellent suggestion, and has parallels with a recent news piece found in *Nature* (Palmer, 2022). WRF-Hydro can indeed be applied elsewhere. The model has been used to study a diverse range of hydrological processes in domains of varying size across the world. Use of WRF-Hydro and choice of spatial resolution is dependent on the existence of requisite boundary conditions, forcing files, and observational constraints. For studies focused on burn scar-debris flow dynamics, the model is again readily adaptable. In our approach, we demonstrate that the burn scar characteristics of a land surface can be set in the land surface model (i.e., reduced canopy height, overland roughness, carboxylation rate, and infiltration rate). Since parameter values will vary based on a myriad of factors (i.e., geography, climate, biome, soil properties, etc.), a major advantage of WRF-Hydro is the ability to modify and calibrate the underlying physical parameters as appropriate for each location. In our revised version of the manuscript, we discussed the application of WRF-Hydro in other regions, and provided a roadmap for global community usage.

In lines 758 – 775, we add a paragraph in the discussion section that reads:

"Lastly, the above discussion of potential WRF-Hydro applications and advancements speaks to the adaptability and customization of this open-source numerical model. An additional layer of WRF-Hydro's

adaptability concerns its geographic focus. While we calibrate and use the model over a central California domain, the choice of geographic footprint is limited only by the availability of requisite initial and boundary conditions, environmental observations for calibration, and computational resources. For use in non-central California domains, we recommend calibration beginning with the default version of the model. Given the ecological and geological diversity of locations that experience wildfires and debris flows, it is likely that calibrations distinct from those reported here will be needed in different regions. For example, soil sealing effects, infiltration, and runoff in wetter and more vegetated locations, such as Oregon, USA, behave differently than those in central California (Palmer, 2022). As such, calibration of relevant model parameters (e.g., saturated hydraulic conductivities) should be based on a physics-informed approach that accounts for local environmental conditions and hydrologic behaviors. Indeed, given the ability to simulate large heterogeneous geographic domains, it is likely that different regions within a given domain may require different calibration schemes. As WRF-Hydro is fully distributed, spatially heterogenous calibrations are non-problematic. This spatial adaptability may prove particularly helpful in post-wildfire debris flow hazard assessments when considering multiple generations of wildfires and variable degrees of burn scar severity and recovery."

The discussion could also compare their model to the USGS model, using a modification to Figure 9, for example, to demonstrate and explain the important differences.

Response: Referee #2 has mentioned a key point that differentiates our susceptibility assessment from the USGS' hazard assessment. That is, the USGS statistical models are able to predict both probability and magnitude of debris flows, which makes them "hazard" assessments, whereas our model is focused on predicting which areas are subject to higher likelihood and should be referred to as "susceptibility" assessment. In the revision, we explained this terminology and altered its usage throughout the manuscript and in the title.

In lines 81 – 90, we explain the difference between "susceptibility" and "hazard": "However, due to long-standing terminology ambiguity in the natural hazard community (Reichenbach et al. 2018), we first begin with a definition of terms. In this study we demonstrate the use of a new physics-based tool to map postfire debris flow susceptibility at regional scales. We follow the guidance of [Reichenbach et al. (2018) & references therein] and define susceptibility as the likelihood of debris flow occurrence in an area, and hazard as the probability of debris flow occurrence of a given magnitude within a specified area and period of time. In other words, debris flow susceptibility does not estimate debris flow size or consider the timing or frequency of the debris flow occurrence. Rather, it focuses on locating areas prone to debris flows considering local environmental factors (Brabb 1985; Guzzetti et al., 2005)."

We also discussed methods that could be employed to create hazard assessments using WRF-Hydro, which would facilitate an apples-to-apples comparison with the USGS product.

In lines 714 – 737: "In addition to investigating the operationalization of WRF-Hydro's natural hazard prediction capabilities, we note that our susceptibility-focused methodology could be advanced to hazard assessment, in line with current USGS products. The USGS Emergency Assessment of Postfire Debris-flow Hazard predicts debris flow volume and likelihood. To advance from susceptibility to hazard assessment, our methodology would need to incorporate both debris flow volume estimates and occurrence likelihoods. In the following, we highlight research directions that could help advance our susceptibility-focused methodological framework. WRF-Hydro is a water-only model. While water-only models have been widely used to investigate and better understand debris flow dynamics (Arattano & Savage, 1994; Tognacca et al., 2000; Arattano & Franzi, 2010; Rengers et al., 2016; McGuire & Youberg, 2020; Di Cristo

et al., 2021), sediment supply, soil erodibility, and other sedimentological factors play important roles in determining the potential for and severity of mass failure events (McGuire et al., 2017). Developing a runoff-generated debris flow model that couples hydrologic and sediment erosion and transport processes could help to characterize postfire debris flow volumes. Indeed, previous efforts have demonstrated the capacity to couple WRF-Hydro with sediment flux models (Yin et al., 2020; Shen et al., 2021). In addition to sediments, burn scar ash can comprise a substantial fraction of the total debris flow volume (e.g., Reneau et al., 2007). As such, efforts to constrain ash availability and entrainment in hydrologic flows could prove fortuitous in hazard assessment and prediction efforts. If WRF-Hydro is not coupled with sediment models, a domain-specific rainfall ID threshold trained with historic landslide inventory and triggering rainfall events (Tognacca et al., 2000; Gregoretti & Dalla Fontana, 2007, 2008) or a newly developed dimensionless discharge and Shields stress threshold (Tang et al., 2019; McGuire & Youberg, 2020) could provide guidance to help identify debris flow triggering time and location, which in turn may improve WRF-Hydro's debris flow initiation identification."

In addition, we discussed methods for probabilistic advancement including systemic investigation of parameter uncertainties and use of ensemble-based precipitation data.

In lines 739 – 755, now it reads: "In addition to constraining potential postfire debris flow volumes, WRF-Hydro's application in debris flow studies could be advanced via concerted engagement with uncertainties that are both external (meteorological forcing data) and internal (physical parameters) to the model. Previous studies have demonstrated that precipitation is often the largest source of uncertainty in hydrologic predictive models (Hapuarachchi et al., 2011; Alfieri et al., 2012). Engagement with precipitation forcing uncertainties in past, near-term, and future contexts could provide probabilistic nuance to natural hazard investigations. For example, (a) debris flow hindcast studies could use a diversity of precipitation datasets to isolate precipitation-derived debris flow uncertainties in historic events, (b) operational forecast efforts could utilize ensemble-based weather forecast data to inform likelihood statements in debris flow hazard assessments, and (c) probabilistic projections of debris flow likelihood in future climates could assess and partition uncertainties derived from emission pathway, model structure, or internal variability effects on meteorological forcings (Nikolopoulos et al., 2019; Hawkins & Sutton, 2009; Deser et al., 2020). Uncertainties internal to WRF-Hydro are also ripe for investigation. Probabilistic predictions crafted from an ensemble of perturbed model physics simulations have been used to predict rainfall-triggered shallow landslides (Raia et al., 2014; Canli et al., 2018; Zhang et al., 2018). Similar efforts using WRF-Hydro could target post-wildfire debris flows."

Specific Comments

Section 5.4 - I don't feel that this is a strong section. It concludes that the hazards are greater in the burned area, and mostly in the channels, and that streamflow is elevated downstream in burned areas, which are not unique findings. Likewise, Figure 11 doesn't come across as strongly as previous figures. I suggest dropping this section.

Response: We somewhat agree with this suggestion. In our revision we removed the regional discussion and Figure 11 from the results portion of the manuscript. However, one of the values added by WRF-Hydro is regional prediction and projection, which differs from more traditional single catchment simulations (e.g., McGuire et al., 2016, 2017). To highlight these capabilities, particularly from a future usage perspective, we added four zoomed-in maps of stream-level and catchment-level debris flow susceptibility over two

other 2020 wildfire burn scars to display more details. we also moved the modified Figure 11 to the discussion section and discussed why regional applications, particularly in an operational setting, provide value.

Revised Figure 11:

[Figure]

**Fig. 11**| MRMS accumulated precipitation and discharge volume informed regional debris flow susceptibility. (a) MRMS accumulated precipitation during January 27th 00:00 to 29th 23:00 over the model domain (shading; mm). Names of burn scars are labeled in black. (b) Accumulated streamflow (yellow-to-red shading; m3) and (c) accumulated overland flow from 27th 00:00 to 28th 12:00 over the model domain (yellow-to-red shading; m3). (d)–(e) Stream-level postfire debris flow susceptibility as Fig. 9b but for River and Camel burn scars. (f)–(g) Catchment-area normalized debris flow susceptibility as Fig. 9e but for River and Camel burn scars. Wildfire perimeters of 2020 wildfire season are outlined in black in (a), in blue in (b), (c), (f), and (g), and in red in (d) and (e). The coastline of California is depicted in grey.

In lines 685 – 702, we discussed why a regional application of WRF-Hydro is important: "In addition to the above results focused primarily on the Dolan burn scar, a key feature of WRF-Hydro is its ability to

simulate the land surface hydrology of expansive geographic domains, e.g., NOAA runs the National Water Model over the entire continental U.S. Development of tools capable of regional susceptibility assessments is crucial, particularly in a wildfire-prone region like California, due to the large spatial scale, diverse morphology, and often tight spatial gradients of precipitation events and their interactions with geographically widespread wildfire burn scars. For example, landfalling ARs are often long (1000s of km) filament-like systems with heterogenous intensity gradients along their length. As a demonstration of wide geographic applicability, we assess susceptibility over our full model domain which includes more than 10,000 catchments and a number of 2020 wildfire burn scars in addition to the Dolan burn scar (Fig 11). The domain-wide analysis reveals elevated discharge volume, i.e., elevated susceptibility, in areas of high precipitation and in burned terrains (Figs. 11a–c). We highlight channelized and catchment-area normalized debris flow susceptibility in non-Dolan burn scar sites in Figs. 11d–g. In an operational forecast context, the ability to simulate landslide and debris flow susceptibilities and hazards over numerous catchments at meteorologically appropriate scales represents a step-change in the field. We argue that our demonstration of WRF-Hydro's debris flow susceptibility hindcast capabilities should motivate further exploration and development for potential use in operational hazard forecasting."

line 489 ff - an interesting note, your modeled discharge increases by 3 or 4 fold matches field measured changes published in Brunkal and Santi for large drainage basins (I could not find the area for your drainage basins, since you include normalized values, but I assume they are more than 5 km^2) (Brunkal, H. and Santi, P., 2017, "Consideration of the Validity of Debris-Flow Bulking Factors," Environmental and Engineering Geoscience, DOI: 10.2113/EEG-1774). See Figure 3 of this paper.

Response: In the revision, we cited this paper in the discussion section to highlight the similarity.

In lines 647 – 652: "A comparison between baseline and burn scar simulations demonstrated that changes in hydraulic properties of burned areas causes drastic changes in surface flows, including faster discharge response times, and greater peak discharge and total volumes, consistent with findings from previous postfire hydrology studies (Anderson et al., 1976; Scott, 1993; Meixner & Wohlgemuth, 2003; Kean et al., 2011; Kinoshita & Hogue, 2015; **Brunkal & Santi, 2016**; Williams et al., 2022)."

Technical Corrections

Figures 1, 7. and 9 could benefit from a bar scale.

Response: In the revision we added scale bars to Figures 1, 7, and 9.

Revised Figure 1:

[Figure]

**Fig. 1**| WRF-Hydro model domain and Dolan burn scar. (a) WRF-Hydro model domain depicting topography, 2020 wildfire season burn scars, and PSL soil moisture and USGS stream gage observing sites. The black rectangle outlines (b) the Dolan burn scar inset, in which debris flow locations and major streams are marked and labeled. The location of the study area is shown in the embedded U.S. map with the state of California shaded in grey.

Revised Figure 7:

[Figure]

**Simulated overland flow and streamflow in burn scar simulation**

**Fig. 7|** WRF-Hydro simulated overland flow and streamflow in the burn scar simulation. (a)–(d) Total volume of accumulated overland flow (yellow-red shading) and streamflow (blue shading) between January 27th 00:00 and 28th 12:00 at four debris flow sites draped over a hillshade of topography. Black rectangles correspond to domains in Fig. 3a–d. Black circles and triangles indicate debris flow source areas and deposits, respectively.

Revised Figure 9:

**Fig. 9**| Discharge volume-based postfire debris flow susceptibility. Debris flow susceptibility at individual stream level for the (a) baseline, (b) burn scar, and (c) difference between burn scar and baseline simulations. Susceptibility is estimated as total discharge volume from January 27th 00:00 to 28th 12:00. (d)–(f) Normalized debris flow susceptibility by catchment area at catchment level. For each catchment, the susceptibility is determined by total discharge volume at the catchment outlet from January 27th 00:00 to 28th 12:00 divided by catchment area.

Figure 9 - the legend is hard to understand. I assume the first bar is volume and the second is normalized volume?

Response: Yes, the first bar is volume and the second is normalized volume. In the revision, we revised the legend to make this clearer. Please see above for the revised Figure 9.

[revised manuscript text omitted]

*Response to referee #2 document for NHESS manuscript 2021-345 by Li et al.*

\*Response is in blue & Sentences or paragraphs after revision are in red

The manuscript deals with assessment of the debris flow hazard in burned areas through simulations that used high-resolution weather radar-derived precipitation. The manuscript has several interesting points, and overall is well written. It is certainly worth to be considered for publication, but I have a couple of points which need to be clarified.

Response: We thank referee #2 for providing valuable comments for the improvement of the manuscript. We have made specific changes in response to each of these comments, and believe that the manuscript has been significantly improved as a result of these changes.

The first (and main) one regards the terminology used. I am afraid that, throughout the article, the term hazard is not used correctly. In my opinion, Authors are rather talking about susceptibility, and not hazard, the difference being that hazard should depict the probability of occurrence of a certain phenomenon not only spatially but also temporally. This latter issue (time) is not considered in the study. I suggest go back to the original definition by Varnes (1984) and UNESCO, and in later works as well, to clarify the meaning of susceptibility and hazard, and to change accordingly the terms in the manuscript.

Response: We thank the referee for highlighting the issue of our loose use of terminology, and we agree with the referee's assessment that our manuscript and methodology are primarily focused on susceptibility, rather than probabilistic hazard assessment. In our revision, we adhere to the definition of terms summarized in Section 2.1 of Reichenbach et al. (2018), *A review of statistically-based landslide susceptibility models*, Earth-Science Reviews. Accordingly, we changed the word "hazard" to "susceptibility" throughout the manuscript and in the title, and we added a paragraph in the introduction section to define the terms.

In lines 81 – 90, now it reads: "Due to this increasing threat, the development of tools to assess postfire debris flow susceptibility and hazards is critical. However, due to long-standing terminology ambiguity in the natural hazard community (Reichenbach et al. 2018), we first begin with a definition of terms. In this study we demonstrate the use of a new physics-based tool to map postfire debris flow susceptibility at regional scales. We follow the guidance of [Reichenbach et al. (2018) & references therein] and define susceptibility as the likelihood of debris flow occurrence in an area, and hazard as the probability of debris flow occurrence of a given magnitude within a specified area and period of time. In other words, debris flow susceptibility does not estimate debris flow size or consider the timing or frequency of the debris flow occurrence. Rather, it focuses on locating areas prone to debris flows considering local environmental factors (Brabb 1985; Guzzetti et al., 2005)."

We added a paragraph to the discussion section which discusses methods by which WRF-Hydro could move beyond "susceptibility" assessments to probabilistic "hazard" quantification. Firstly, we can estimate debris flow volume by coupling WRF-Hydro with a sediment erosion and transport model, which has been proved successful in previous studies (Yin et al., 2020; Shen et al., 2021). If WRF-Hydro is not coupled with any sediment models, rainfall intensity-duration thresholds (Tognacca et al., 2000; Gregoretti & Dalla Fontana, 2007, 2008) or dimensionless discharge and Shields stress thresholds (Tang et al., 2019; McGuire & Youberg, 2020) can provide information on identifying debris flow triggering time and location.

In lines 714 – 737, now it reads: "In addition to investigating the operationalization of WRF-Hydro's natural hazard prediction capabilities, we note that our susceptibility-focused methodology could be advanced to hazard assessment, in line with current USGS products. The USGS Emergency Assessment of Postfire Debris-flow Hazard predicts debris flow volume and likelihood. To advance from susceptibility to hazard assessment, our methodology would need to incorporate both debris flow volume estimates and occurrence

likelihoods. In the following, we highlight research directions that could help advance our susceptibility-focused methodological framework. WRF-Hydro is a water-only model. While water-only models have been widely used to investigate and better understand debris flow dynamics (Arattano & Savage, 1994; Tognacca et al., 2000; Arattano & Franzi, 2010; Rengers et al., 2016; McGuire & Youberg, 2020; Di Cristo et al., 2021), sediment supply, soil erodibility, and other sedimentological factors play important roles in determining the potential for and severity of mass failure events (McGuire et al., 2017). Developing a runoff-generated debris flow model that couples hydrologic and sediment erosion and transport processes could help to characterize postfire debris flow volumes. Indeed, previous efforts have demonstrated the capacity to couple WRF-Hydro with sediment flux models (Yin et al., 2020; Shen et al., 2021). In addition to sediments, burn scar ash can comprise a substantial fraction of the total debris flow volume (e.g., Reneau et al., 2007). As such, efforts to constrain ash availability and entrainment in hydrologic flows could prove fortuitous in hazard assessment and prediction efforts. If WRF-Hydro is not coupled with sediment models, a domain-specific rainfall ID threshold trained with historic landslide inventory and triggering rainfall events (Tognacca et al., 2000; Gregoretti & Dalla Fontana, 2007, 2008) or a newly developed dimensionless discharge and Shields stress threshold (Tang et al., 2019; McGuire & Youberg, 2020) could provide guidance to help identify debris flow triggering time and location, which in turn may improve WRF-Hydro's debris flow initiation identification."

In addition, we also discussed methods for probabilistic advancement including systemic investigation of parameter uncertainties and use of ensemble-based precipitation data.

In lines 739 – 755, now it reads: "In addition to constraining potential postfire debris flow volumes, WRF-Hydro's application in debris flow studies could be advanced via concerted engagement with uncertainties that are both external (meteorological forcing data) and internal (physical parameters) to the model. Previous studies have demonstrated that precipitation is often the largest source of uncertainty in hydrologic predictive models (Hapuarachchi et al., 2011; Alfieri et al., 2012). Engagement with precipitation forcing uncertainties in past, near-term, and future contexts could provide probabilistic nuance to natural hazard investigations. For example, (a) debris flow hindcast studies could use a diversity of precipitation datasets to isolate precipitation-derived debris flow uncertainties in historic events, (b) operational forecast efforts could utilize ensemble-based weather forecast data to inform likelihood statements in debris flow hazard assessments, and (c) probabilistic projections of debris flow likelihood in future climates could assess and partition uncertainties derived from emission pathway, model structure, or internal variability effects on meteorological forcings (Nikolopoulos et al., 2019; Hawkins & Sutton, 2009; Deser et al., 2020). Uncertainties internal to WRF-Hydro are also ripe for investigation. Probabilistic predictions crafted from an ensemble of perturbed model physics simulations have been used to predict rainfall-triggered shallow landslides (Raia et al., 2014; Canli et al., 2018; Zhang et al., 2018). Similar efforts using WRF-Hydro could target post-wildfire debris flows."

Another point which needs more details is the description of the debris flows. Authors talk about several debris flows that occurred, and start to cite them in section 2.1. However, a clear description of the events, in terms of geology, morphology, morphometry, volumes is never properly given. This should be done the first time debris flows are mentioned (possibly in section 2.1) to let the reader understand the main characters of the events. For instance, were these debris flows individual phenomena, or did they start from multiple source areas? Further, were they channelized or openslope? More geomorphological info would be useful to understand the conditions under which the debris flows initiated and developed. Only at page 18 some info are provided, but these should appear much before than that, and be well organized, rather than distributed in different parts of the manuscript.

Response: We sympathize with the referee's desire for more data on the debris flows highlighted in our manuscript. However, at present, there have been no systematic studies of these debris flows, so while we are quite confident that debris flows occurred at these locations based on field observations of the deposits and our remote sensing analyses, information about source areas (which are in extremely inaccessible locations) and volumes are not well constrained. We do know that David Cavagnaro et al. (https://agu.confex.com/agu/fm21/meetingapp.cgi/Paper/921613) have undertaken a huge effort to map debris flows in the Dolan Fire burn scar. I suspect this forthcoming work will be able to answer some of the referee's concerns in greater detail.

In our revised manuscript we added a paragraph to section 2.2 to describe the debris flow geologic setting. We moved the descriptions on the debris flows in section 5.2 to section 2.2. Our information on the geology come from the USGS geologic map (https://mrdata.usgs.gov/geology/state/state.php?state=CA). We also calculated mean and maximum slope values for the triggering and deposition sites of the four debris flow events using the slope calculation function in ArcMap and the USGS 30-m DEM.

In lines 235 – 262, it now reads:

**"2.2 Debris flow geologic setting**

According to the USGS National Elevation Dataset 30-m digital elevation model (DEM), the Rat Creek debris flow sits at the base of a 1st order catchment with a drainage area of 2.23 km$^2$. Mill Creek, Big Creek, and Nacimiento debris flows were initiated within extremely steep, intensely burned, 1st order catchments, but were deposited in 2nd, 3rd, and 3rd Strahler stream order channels, respectively. All four debris flows were channelized. Rat Creek, Mill Creek, and Big Creek debris flow deposition sites have elevations ranging from 20–60 m, while Nacimiento debris flow deposited at an elevation of ~440 m above sea level. We calculate catchment slopes using the DEM and the slope calculation function in ArcMap. The average slope of the catchments containing Rat Creek and Mill Creek debris flow deposition sites is ~25°. The average catchment slope of Big Creek deposition site is ~28° and Nacimiento is ~21°. For debris flow source areas, the average and maximum slopes of Mill Creek are 23° and 39°, 21° and 43° for Big Creek, and 24° and 41° for Nacimiento. According to the Soil Survey Geographic Database and California geologic map data, surface soils at the three coastal debris flow sites (i.e., Rat Creek, Mill Creek, and Big Creek) are texturally classified as loam with underlying Franciscan Complex sedimentary rocks of Jurassic to Cretaceous age. Soil at Nacimiento is classified as sandy loam with underlying Upper Cretaceous and Paleocene marine sedimentary rocks from the Dip Creek Formation, Asuncion Group, Shut-In Formation, Italian Flat Formation, Steve Creek Formation, and El Piojo Formation. Mill Creek, Big Creek, and Nacimiento were relatively large debris flows with runout lengths between ~2–5 km, while Rat Creek occurred in a smaller catchment and had a runout length of ~300 m. The difference in runout length and debris flow size is primarily controlled by upstream catchment size, however for the three coastal debris flow events at Rat Creek, Big Creek, and Mill Creek, also constrained by the downslope ocean. We note that there were likely more debris flows triggered during the AR event. The four debris flow events highlighted here were identified during brief post-event field excursions due to their intersection with major roadways. Given that our primary goal here is to demonstrate the utility of WRF-Hydro – a comprehensive catalogue of debris flows is beyond the scope of this study, although underway by other researchers (Cavagnaro et al., 2021)."

Other issues:

Figure 1 definitely needs a location map, showing where we are in California, and in USA. Authors give for granted that anybody knows the site, but for an international journal a location map is always necessary.

Response: In our revision we added a location map of the USA which depicts the locations of California and the burn scar/debris flow region.

Revised Figure 1:

[Figure]

**Fig. 1**| WRF-Hydro model domain and Dolan burn scar. (a) WRF-Hydro model domain depicting topography, 2020 wildfire season burn scars, and PSL soil moisture and USGS stream gage observing sites. The black rectangle outlines (b) the Dolan burn scar inset, in which debris flow locations and major streams are marked and labeled. The location of the study area is shown in the embedded U.S. map with the state of California shaded in grey.

Throughout the manuscript, references should be listed in chronological order when more than two references are cited. Some incomplete or wrong references are present in the list. Please check at this regard the attached file. Eventually, some minor issues are indicated in the accompanying file.

Response: In our revision we reordered the in-text citations chronologically and corrected the references. We thank referee #2 for their very careful examination of references. We have checked the attached file and addressed those minor issues accordingly.

Overall, I evaluate positively the manuscript, which however needs to clarify the points outlined above, and recommend minor revisions.

We thank referee #2 again for their careful review and positive comments.

**References:**

[revised manuscript text omitted]

---

## Referee Report (RR1)

**Peer review report on "Augmentation of WRF-Hydro to Simulate Overland Flow- and Streamflow-Generated Debris Flow Susceptibility in Burn Scars" (nhess-2021-345)**

**General Comments**

This paper presents the hydrologic analysis in post-fire debris flow by employing WRF-Hydro. The research extends an important hydrologic modeling tool to an important topic, post-fire hydrologic hazards. The paper is mainly an application of WRF-Hydro over a large area including burned scars in CA. It integrates multiple techniques and datasets, such as hydrologic simulations, radar data, and debris flow identification using RS, and so on, to improve hydrologic analysis. Also, the revised manuscript has big improvement. It definitely merits publication after consideration of these comments.

One problem with this research is that the debris flow susceptibility (or likelihood) is indicated using water volume. Runoff-generated debris flow occurs when flow strengths exceed the threshold, authors also noticed and mentioned in the manuscript. The key to debris flow occurrence is flow strengths, such peak flow, maximum depth, maximum velocity and so on, rather than volume. Of course, generated volume is closely related to peak flow but also controlled by duration.

A related problem is the spatial resolution of computational cell. The 100m grid is used in the study. It is okay to simulate the overland flow generation and hydrograph at the outlets, but it provides few information about flow dynamics at fine resolution, which is the key to observe the debris flow generation and occurrence.

A highlight of this study is output overland flow by modifying the source code. The overland flow generation is represented by Noah-MP. The related variables are $q\_sfcflx\_x$ and $q\_sfcflx\_y$. and combined with $qqsfc$ and $Noah\_distr\_routing$ to calculate the amount (forgive me if I remember wrong). It is very valuable to point out the modification. I believe the modification of the source code will be published with this work. It will benefit WRF-Hydro community!

Another problem is "atmospheric river" (AR) mentioned many times, and it seems authors emphasize the AR is the major reason of heavy rainstorm of post-fire debris flow. Generally, it is fine. But AR does not the directly produce rainfall and AR is a very large scale atmospheric pattern occurred many places. AR-triggered certain synoptic system, such as NCFR carrying plenty of moisture provides opportunities of heavy rainfall.

The third problem is the performance of MRMS data in the study area. I believe MRMS data is the best option author may have, but its poor quality in mountainous area always unable catch the intensive storms. I'd like to see how authors consider this issue.

A similar problem is only three stream gauges are used to calibrate the model for such a large area. One reason might be lack of natural flow record. You may want to use natural flow data, such as https://rivers.codefornature.org/, https://pubs.er.usgs.gov/publication/70046617

Temporal resolution may be different but still provide valuable information.

**Detailed Comments**

**Line 151.** Spatial resolution of 100m is problematic for capturing debris flow behavior. This comment applies to the same issue throughout the text (e.g., lines 273, 332, etc.).

**Line 153-154**. Other studies did use water-only models but those modeling works were done by using high resolution DEM, such as 1-m lidar data. The flow dynamics at fine resolution describes the initiation of debris flow (e.g., Rengers et al., 2016; Mcguire et al., 2017)

Rengers, F.K., McGuire, L.A., Kean, J.W., Staley, D.M. and Hobley, D.E.J., 2016. Model simulations of flood and debris flow timing in steep catchments after wildfire. Water Resources Research, 52(8), pp.6041-6061.
McGuire, L. A., Rengers, F. K., Kean, J. W., and Staley, D. M. (2017), Debris flow initiation by runoff in a recently burned basin: Is grain-by-grain sediment bulking or en masse failure to blame?, Geophys. Res. Lett., 44, 7310– 7319, doi:10.1002/2017GL074243.

**Line 179**. Not clear. You may want to point out the time window of rainfall intensity, such as 15-min or 30-min.

**Line 192**. Add spatial resolution in this section.

**Lines 223-225.** You may add a figure to compare the RS-based debris flow and field observation.

**Line 262**. Simplifying the model description? WRF-Hydro has been used extensively.

**Line 369.** Justify 3 gauges used for calibration is sound. Or you may consider to use natural flow data:

https://rivers.codefornature.org/, https://pubs.er.usgs.gov/publication/70046617, or https://pubs.er.usgs.gov/publication/70046617

**Line 376.** Justify 1-hour results can represent the flow peaks.

**Line 453,455**. Change unit of Ks to mm/hr.

**Line 505-506.** Great metrics you used here in the context of hazards and their impacts.

**Line 563-565.** I suspect the accumulated volume is a reasonable metrics for runoff-generated debris flow assessment. Initiation of runoff-generated debris flow is a threshold behavior, which means the peak flow or unitless peak flow must be larger than certain threshold, or it is unlikely to happen. You may want to look at this paper and references therein: https://agupubs.onlinelibrary.wiley.com/doi/pdf/10.1029/2019GL083623?msclkid=28d5425bbaed11ecaab0029541ed7fcc

**Line 669-672.** I agree the uniform precip introduce bias of hazards distribution over watersheds. But I highly suspect the assessments based on the data and method in this study, such as the computation grid size, MRMS quality in the study area, parameters on burned scars, and metrics (water volume rather than peak flow) used for debris flow assessment.

**Line 730-736.** You got the idea but confused why the volume of water is used for debris flow assessment.

**Figure 4:** change rainrate unit to mm/hr, which is more common for post-fire hydrologic study.

You may tune font size in all figures.

---

## Author Response (AR2)

We thank the referee for their insightful comments. We have made constructive changes in response to each of these comments and believe that the manuscript is substantially improved as a result.

This Response to the Referee document provides a complete documentation of the changes that are responsive to each Referee Comment. The document is designed so that the changes that we have made in response to each comment can be immediately read and understood. Referee Comments are shown in black, while Author Responses are shown in blue. Quoted text from the revised manuscript is shown in red. For clarity, we group our responses to the referee's thematically similar General and Specific comments into individual numbered responses below.

**Referee #3:**

This paper presents the hydrologic analysis in post-fire debris flow by employing WRF-Hydro. The research extends an important hydrologic modeling tool to an important topic, post-fire hydrologic hazards. The paper is mainly an application of WRF-Hydro over a large area including burned scars in CA. It integrates multiple techniques and datasets, such as hydrologic simulations, radar data, and debris flow identification using RS, and so on, to improve hydrologic analysis. Also, the revised manuscript has big improvement. It definitely merits publication after consideration of these comments.

**Response:** We thank the referee for their careful review and positive assessment of our manuscript. In the below, we address each of the referee's comments. We hope that the changes have helped improve the clarity of the manuscript and increased the insights gained from the results. We thank the referee for judging this manuscript suitable for publication, and hope that our revision efforts further that sentiment.

**1. Peak Flow Comments**

**General Comments:** One problem with this research is that the debris flow susceptibility (or likelihood) is indicated using water volume. Runoff-generated debris flow occurs when flow strengths exceed the threshold, authors also noticed and mentioned in the manuscript. The key to debris flow occurrence is flow strengths, such peak flow, maximum depth, maximum velocity and so on, rather than volume. Of course, generated volume is closely related to peak flow but also controlled by duration.

**Specific Comments:**
**Line 563-565.** I suspect the accumulated volume is a reasonable metrics for runoff-generated debris flow assessment. Initiation of runoff-generated debris flow is a threshold behavior, which means the peak flow or unitless peak flow must be larger than certain threshold, or it is unlikely to happen. You may want to look at this paper and references therein:
https://agupubs.onlinelibrary.wiley.com/doi/pdf/10.1029/2019GL083623?msclkid=28d5425bbaed11ecaab0029541ed7fcc

**Response #1:** We thank the referee for their feedback regarding the use of accumulated volume in our susceptibility analysis, however we are confused by the contradictory nature of the above two comments. In the first comment the referee indicates that the use of accumulated volume may be a "problem", while in the second comment they indicate their suspicion that "the accumulated volume is a reasonable metric." Despite the contradictory nature of these comments, in the following, we assume the referee is seeking greater clarity on the use of accumulated volume versus peak discharge in debris flow susceptibility assessments.

It is important to note that in our manuscript we do not attempt to simulate debris flow initiation or the dynamics of debris flows. In this manuscript, we restrict our analysis to the assessment of debris flow susceptibility by locating areas with elevated environmental conditions conducive to debris flow occurrence (Brabb 1985; Guzzetti et al., 2005). We agree with the referee that if we were to attempt to simulate the dynamics of individual debris flows (e.g., initiation, entrainment, runout, etc.) rather than assess susceptibility, a different modeling framework would be needed [e.g., the individual catchment-level framework used by Rengers et al. (2016) and McGuire et al. (2017)]. Indeed, to resolve debris flow initiation requires spatiotemporal resolutions of meters and seconds that are not conducive to regional analyses.

To clarify the goal of this study, we revised the sentence in lines 96 – 98, which now reads (the edits we made is underscored): In other words, debris flow susceptibility neither simulates debris flow dynamics such as initiation nor estimates debris flow size or considers the timing or frequency of the debris flow occurrence. Rather, it focuses on locating areas prone to debris flows considering local environmental factors (Brabb 1985; Guzzetti et al., 2005).

We are in agreement with the referee that accumulated volume is closely related to peak discharge. Indeed, in our susceptibility analysis, we find similar susceptibility results regardless of variable choice. However, we concede that in the mechanistic debris flow world, peak discharge is likely a more valuable metric to quantify. As such, we have substantially revised our manuscript to now assess debris flow susceptibility using peak discharge output from WRF-Hydro. In addition, we have altered all related text, figures, and tables accordingly. Below, we provide the new peak flow versions of Figures 9, 10, and 11 and Table 3 for your reference.

In our revision, we've moved all accumulated discharge volume figures and tables to Appendix B (Figs. B7–9 and Table B5). For the convenience of comparison, we also reproduce accumulated discharge volume figures below (Fig. B7). For this domain under these meteorological conditions we generally find that similar conclusions regarding susceptibility can be drawn using either peak discharge or accumulated discharge volume, i.e., Rat Creek had medium susceptibility, Mill Creek had high susceptibility, and Big Creek and Nacimiento had very high susceptibility in the burn scar simulation, and catchments with catchment-area normalized peak discharge correspond well with the post-event debris flow identification (Figs. 9b,e & B7b,e). However, it is in the difference plots that the value of a peak discharge-based analysis is apparent. That is, streams and catchments with elevated susceptibility are more evident in the peak discharge maps (Figs. 9c,f & B7c,f). We thank the referee for this suggestion and hope that our substantial changes better highlight our methodological capabilities and results.

**Postfire debris flow susceptibility**

[Figure]

**Catchment-area normalized postfire debris flow susceptibility**

[revised manuscript text omitted]

**2. Spatial and Temporal Resolution Comments**

**General Comments:**
A related problem is the spatial resolution of computational cell. The 100m grid is used in the study. It is okay to simulate the overland flow generation and hydrograph at the outlets, but it provides few information about flow dynamics at fine resolution, which is the key to observe the debris flow generation and occurrence.

**Specific Comments:**
**Line 151.** Spatial resolution of 100m is problematic for capturing debris flow behavior. This comment applies to the same issue throughout the text (e.g., lines 273, 332, etc.).
**Line 376.** Justify 1-hour results can represent the flow peaks.

**Response #2:** We thank the reviewer for this question and an opportunity for us to clarify (a) the objectives of our paper and (b) the spatial and temporal resolution of our different modeling components:

(a) As mentioned in Response #1, it is important to note that in our manuscript we do not attempt to simulate debris flow initiation or the dynamics of debris flows. In this manuscript, we restrict our analysis to the assessment of debris flow susceptibility by locating areas with elevated environmental conditions conducive to debris flow occurrence. We agree with the referee that if we were to attempt to simulate the dynamics of individual debris flows (e.g., initiation, entrainment, runout, etc.) rather than assess susceptibility, a different modeling framework would be needed [e.g., the individual catchment-level framework used by Rengers et al. (2016) and McGuire et al. (2017)]. Indeed, to resolve debris flow initiation requires spatiotemporal resolutions that are not conducive to regional analyses.

(b) To clarify the spatial resolutions used in our study, here we describe the various components of our modeling scheme. Channelized streamflow is simulated by the channel routing module of WRF-Hydro at fine spatial resolutions that range from 1.5 to 100 m depending on the USGS DEM-indicated stream order. This information can be found in Figure B3 and Table B1 in our manuscript. Since our model domain is quite large (~35,000 km$^2$), streamflow is then output on the 100-m grid due to limited storage space. However, discharge dynamics are solved at a resolution consistent with the stream order.

As for temporal resolution, the MRMS precipitation forcing is hourly but our WRF-Hydro terrain routing and channel routing modules compute overland flow and channelized streamflow every 6 seconds. However, we output simulated data at an hourly resolution due to limited storage space.

To clarify our methodology, we added a sentence to lines 288 – 293, lines 328 – 331, and lines 356 – 361 in our revised manuscript (the added sentence is underscored). Lines 288 – 293 now read: The channel routing module then calculates channelized flows assuming a trapezoidal channel shape (Fig. B2). Parameters related to the trapezoidal channel, such as channel bottom width ($B_w$), Manning's roughness coefficient (n), and channel side slope (z) are functions of channel stream order (Fig. B3 and Table B1). Channelized streamflow is computed at spatial resolutions ranging from 1.5 m to 100 m depending on the channel stream order (Table B1). Computed streamflow is then output on the 100-m grid.

Lines 328 – 331 now read (the added sentence is underscored): Noah-MP passes excess water to the terrain routing module, which simulates overland flow using a 2-dimensional fully-unsteady, explicit, finite-difference diffusive wave equation adapted from Julien et al. (1995) and Ogden (1997). In this application, overland flow is computed at each 6 second time step and is archived hourly at 100-m spatial resolution.

Lines 356 – 361 now read (the added sentence is underscored): If overland flow intersects grid cells identified as channel grids (2nd Strahler stream order and above; pre-defined by the hydrologically conditioned USGS 30-m DEM), the channel routing module routes the water as channelized streamflow using a 1-dimensional, explicit, variable time-stepping diffusive wave formulation. In this work, the channel routing module calculates streamflow at 6-s temporal resolution and spatial resolutions ranging from 1.5 m to 100 m depending on the channel stream order (Fig. B3 and Table B1).

**3. MRMS Data**

**General Comments:** The third problem is the performance of MRMS data in the study area. I believe MRMS data is the best option author may have, but its poor quality in mountainous area always unable catch the intensive storms. I'd like to see how authors consider this issue.

**Response #3:** We thank the referee for pointing out the contingencies with using MRMS data, and we agree that given all available options, it is the "best choice." MRMS data has relatively high

spatial (1-km) and temporal resolutions (hourly) compared to other available datasets. In addition, it covers a large spatial domain (i.e., CONUS) and thus it is valuable for regional susceptibility assessments. We acknowledge the uncertainties in the MRMS precipitation data on lines 701 – 706 and in Appendix A. We note that MRMS Gauge-Corrected and MRMS Mountain Mapper Precipitation datasets may be superior products and are therefore preferred in mountainous areas, however these datasets are not yet available for our study period (Jan 1–31, 2021) (https://mtarchive.geol.iastate.edu/2021/01/31/mrms/ncep/; retrieved May 2022). In addition, it is important to note that from an operational forecast application perspective, predicted precipitation products will suffer similar (and likely worse) limitations to that of MRMS. It is therefore important to work with the best data resources available. We agree with the referee that the best available data set is MRMS.

A discussion on this topic can be found in the manuscript on lines 701 – 706: However, this also means the accuracy of WRF-Hydro predictions depends on the accuracy of precipitation forcing, and in our hindcast application, MRMS precipitation data (Appendix A). Accordingly, our WRF-Hydro-based assessment could benefit from precipitation products mosaiced from various sources to constrain precipitation-based uncertainties (e.g., gauge-corrected and/or Mountain Mapper MRMS), although the long processing time of these datasets inhibits timely post-event assessments.

We have also added a sentence in Appendix A to highlight the caveats of using MRMS in mountainous areas. Appendix A now reads (the changes we made are underscored):

**Appendix A**

**Text A1. Multi-Radar/Multi-Sensor System (MRMS) radar-only precipitation estimate and uncertainty**

MRMS is a precipitation product that covers the contiguous United States (CONUS) on 1-km grids. It combines precipitation estimates from sensors and observational networks (Zhang et al., 2011, 2014, 2016), and is produced at the National Centers for Environmental Prediction (NCEP) and distributed to National Weather Service forecast offices and other agencies. Input datasets used to produce MRMS include the U.S. Weather Surveillance Radar-1988 Doppler (WSR-88D) network and Canadian radar network, Parameter-elevation Regressions on Independent Slopes Model (PRISM; Daly et al., 1994, 2017), Hydrometeorological Automated Data System (HADS) gauge data with quality control (Qi et al., 2016), and outputs from numerical weather prediction models. There are four different MRMS quantitative precipitation estimates (QPE) products incorporating different input data or combinations: radar only, gauge only, gauge-adjusted radar, and Mountain Mapper. One limitation of using MRMS radar only precipitation data is that radars struggle to capture rainfall in mountainous areas due to orographic beam blocking (Anagnostou et al. 2010; Germann et al. 2007). Gauge-corrected and Mountain Mapper MRMS are thought to be superior products in mountainous terrains and therefore are preferred. However, for our study period (i.e., January 1–31, 2021), the gauge-corrected and Mountain Mapper MRMS are not available (as of May 2022).

We acknowledge that precipitation data has uncertainties. Use of different precipitation products may help to constrain uncertainties. A study comparing different gridded precipitation datasets including satellite-based precipitation data, gauge dataset, and multi-sensor products

revealed large uncertainties in precipitation intensity (Bytheway et al., 2020). However, comparing different precipitation datasets to characterize uncertainties is beyond the scope of this study. MRMS provides gridded precipitation at high temporal (hourly) and spatial (1-km) resolutions, making it a useful tool to demonstrate the utility of WRF-Hydro in post-wildfire debris flow susceptibility assessments.

**4. Overland Flow Augmentation**

**General Comments:** A highlight of this study is output overland flow by modifying the source code. The overland flow generation is represented by Noah-MP. The related variables are *q_sfcflx_x* and *q_sfcflx_y*. and combined with *qqsfc* and *Noah_distr_routing* to calculate the amount (forgive me if I remember wrong). It is very valuable to point out the modification. I believe the modification of the source code will be published with this work. It will benefit WRF-Hydro community!

**Response #4:** We agree and thank the reviewer characterizing our contribution as very valuable. We hope so! We provide a link to our modified code in the *Code availability statement* at the end of the manuscript (lines 1018 – 1020). At this link we provide the modified source code and instructions on how to use the modified Fortran files. In our revision, we've added a sentence to point readers to the *Code availability statement* on lines 349 – 351.

Lines 349 – 351 now read (the sentence we added is underscored): One key advance made in this work is that we modified WRF-Hydro source code to output overland flow (see the Code availability statement for the modified source code).

**5. Atmospheric Rivers**

**General Comments:** Another problem is "atmospheric river" (AR) mentioned many times, and it seems authors emphasize the AR is the major reason of heavy rainstorm of post-fire debris flow. Generally, it is fine. But AR does not the directly produce rainfall and AR is a very large scale atmospheric pattern occurred many places. AR-triggered certain synoptic system, such as NCFR carrying plenty of moisture provides opportunities of heavy rainfall.

**Response #5:** We thank the referee for highlighting the confusion produced by our manuscript regarding atmospheric rivers. According to the glossary of the American Meteorological Society (https://glossary.ametsoc.org/wiki/Atmospheric_river), an atmospheric river (AR) is defined as "a long, narrow, and transient corridor of strong horizontal water vapor transport" that "frequently leads to heavy precipitation where they are forced upward—for example, by mountains or by ascent in the warm conveyor belt.".

We agree that ARs are large synoptic systems, which partly motivates our use of a regional hydrologic model to study debris flow susceptibility, that is, an individual landfalling AR will alter debris flow likelihood over a large region (the area of an AR is typically on the order of $10^5$ km$^2$), rather than within an individual catchment. Indeed, ARs are reported to produce 30–50% of the annual precipitation and 60%–100% of the extreme precipitation along the U.S. west coast (Collow et al., 2020; Eldardiry et al., 2019; Hecht & Cordeira, 2017). Given their importance to

hydrological conditions over our modeling domain, we have added text to lines 75 – 88 that defines ARs and contextualizes their importance:

Lines 75 – 88 now read: On the U.S. west coast, atmospheric rivers (ARs) are the dominant synoptic weather systems responsible for producing postfire debris flows (Barth et al., 2017; Oakley et al., 2017, 2018; Young et al., 2017). ARs are long filament-like bands of elevated water vapor within the lower troposphere that often form over ocean basins. They are responsible for over 90% of poleward water vapor transport (Zhu & Newell, 1998) and often result in heavy precipitation upon landfall, particularly with orographic uplift (Ralph et al., 2004; Neiman et al., 2008). It is reported that 30–50% of annual precipitation and 60%–100% of extreme precipitation along the U.S. west coast is the result of ARs (Collow et al., 2020; Eldardiry et al., 2019; Hecht & Cordeira, 2017). In California, anthropogenic climate change is projected to increase AR intensity (Huang et al., 2020a, 2020b), increase the intensity and frequency of wet-season precipitation (Polade et al., 2017; Swain et al., 2018), increase wildfire potential (Brown et al., 2020; Swain 2021), and extend the wildfire season (Goss et al., 2020). As such, the occurrence and intensity of postfire debris flows are likely to increase as the effects of anthropogenic climate change persist (Cannon & DeGraff, 2009; Kean & Staley, 2021; Oakley 2021).

**6. Calibration**
**General Comments:** A similar problem is only three stream gauges are used to calibrate the model for such a large area. One reason might be lack of natural flow record. You may want to use natural flow data, such as https://rivers.codefornature.org/, https://pubs.er.usgs.gov/publication/70046617 Temporal resolution may be different but still provide valuable information.

**Specific Comments:**
**Line 369.** Justify 3 gauges used for calibration is sound. Or you may consider to use natural flow data:
https://rivers.codefornature.org/,          https://pubs.er.usgs.gov/publication/70046617,          or https://pubs.er.usgs.gov/publication/70046617

**Response #6:** We thank the reviewer for providing extra data sources. Indeed, records of natural flows are scarce in California. We chose the three USGS stream gages because they are located downstream of Dolan burn scar which is the focus of the case study.

For reference, lines 385 – 393 of our manuscript read: Due to the Mediterranean climate of California, many USGS stream gages experience low or no flow during the dry season. In addition, many gages are under manual regulation to mitigate wet-season flood risks and better distribute water resources. As such, it can be challenging to obtain natural streamflow observations for model calibration. Here, three USGS stream gages [i.e., Arroyo Seco NR Greenfield, CA (ID 11151870), Arroyo Seco NR Soledad, CA (ID 11152000), and Arroyo Seco BL Reliz C NR Soledad, CA (ID 11152050)] (Fig. 1a) on streams that have measurable flows during our study period and are free of human regulation are used. These gages are located downstream of the Dolan burn scar and hence are useful in calibrating the parameters associated with burn scar effects.

We investigated the two links provided by the reviewer. The first suggested dataset (https://rivers.codefornature.org/) provides machine-learning estimates of natural flows at monthly

resolution. Their methodology is provided at this link: https://rivers.codefornature.org/#/science. In line with the referee's opinion that coarse temporal resolutions are not ideal for debris flow assessments, we do not think monthly flow records are helpful in calibrating the model to assess debris flow susceptibility.

The second dataset the reviewer suggested is the USGS GAGESII data (https://pubs.er.usgs.gov/publication/70046617). However, this is not a streamflow time series data. Instead, it provides geospatial data and classifications for the USGS stream gages including shapefiles of the point locations, basins, and streamlines of the USGS stream gages. After carefully comparing the point locations of GAGESII with the locations of USGS stream gages which we are already using (https://maps.waterdata.usgs.gov/mapper/?state=ca), we found that they closely match. As a result, the above two datasets do not provide additional useable information.

**Detailed Comments**

**Line 153-154**. Other studies did use water-only models but those modeling works were done by using high resolution DEM, such as 1-m lidar data. The flow dynamics at fine resolution describes the initiation of debris flow (e.g., Rengers et al., 2016; Mcguire et al., 2017)
Rengers, F.K., McGuire, L.A., Kean, J.W., Staley, D.M. and Hobley, D.E.J., 2016. Model simulations of flood and debris flow timing in steep catchments after wildfire. Water Resources Research, 52(8), pp.6041-6061.
McGuire, L. A., Rengers, F. K., Kean, J. W., and Staley, D. M. (2017), Debris flow initiation by runoff in a recently burned basin: Is grain-by-grain sediment bulking or en masse failure to blame?, Geophys. Res. Lett., 44, 7310– 7319, doi:10.1002/2017GL074243.

**Response #7:** We agree with the reviewer that these studies simulated debris flow initiation at a high resolution. However, we are not attempting to do what these studies have done. We are assessing debris flow susceptibility over a regional domain. We have balanced the computational costs between fine spatiotemporal resolution and large spatial extent. Our model domain covers more than 10,000 catchments and 35,000 $km^2$, while the above mentioned mechanistic studies are focused on individual catchments [the study area is 0.01 $km^2$ in McGuire et al. (2017) and <2 $km^2$ in Rengers et al. (2016)].

To avoid the confusion regarding our intent, we deleted the word "behavior" from the following sentence:

The sentence in lines 164 – 168 of our revised manuscript now reads: Previous efforts, employing shallow water equations, diffusive, kinematic, and diffusive-kinematic wave models, have demonstrated that water-only models can provide critical insights on runoff-driven debris flows (Arattano & Savage, 1994; Arattano & Franzi, 2010; Di Cristo et al., 2021), even in burned watersheds (Rengers et al., 2016; McGuire & Youberg, 2020).

And given the lack of clarity in our original manuscript, we have modified the text in the Introduction Section that discusses these studies.

Lines 130 – 141 now read: Studies that have investigated postfire hydrologic responses using physics-based models have largely focused on mechanistic studies such as short-term responses at high spatiotemporal resolutions (Rengers et al., 2016; McGuire et al., 2016, 2017) or long-term runoff responses at coarse temporal resolutions (McMichael & Hope, 2007; Rulli & Rosso, 2007) in individual catchments. For example, process-based models have employed shallow water equations to better understand the triggering (McGuire et al., 2017; Tang et al., 2019a, 2019b) and sediment transport mechanisms (McGuire et al., 2016) of postfire debris flows as well as the timing of postfire debris flows (Rengers et al., 2016). The numerical models employed by these studies are used to simulate debris flow dynamics rather than assess susceptibility over regional domains, as such they focus on individual catchments (with drainage areas of ~1 km2) with very high spatiotemporal resolutions (Rengers et al., 2016; McGuire et al., 2016, 2017; Tang et al., 2019a, 2019b).

**Line 179**. Not clear. You may want to point out the time window of rainfall intensity, such as 15-min or 30-min.

**Response #8:** To clarify, we have modified this sentence to indicate that the precipitation data we used to calculate the 24 mm hr$^{-1}$ is hourly MRMS.

The sentence now reads: On January 27–29, 2021, an atmospheric river (AR) made landfall on the Big Sur coast, bringing more than 300 mm of rainfall to California's Coast Ranges (Fig. 2), with a peak rainfall rate of 24 mm h$^{-1}$ [calculated with Multi-Radar/Multi-Sensor System (MRMS) precipitation; Zhang et al., 2011, 2014, 2016].

**Line 192**. Add spatial resolution in this section.

**Response #9:** The spatial resolution of Sentinel-2 optical data that is used to calculate rdNDVI is 10 m. The SAR-change is calculated from the Sentinel-1 satellites which also have a spatial resolution of 10 m.

In our revised manuscript, we added the spatial resolution to lines 215 – 217. It now reads (the changes are underscored): …where NIR is the near-infrared response and Red is the visible red response. rdNDVI was calculated from 10-m Sentinel-2 satellite data using the HazMapper v1.0 Google Earth Engine application (Scheip & Wegmann, 2021).

And lines 224 – 226: Lastly, we searched for debris flows (and other ground surface deformation) by examining SAR backscatter change with data acquired by the 10-m Copernicus Sentinel-1 (S1) satellites [see full description in Handwerger et al. (2022)].

**Lines 223-225.** You may add a figure to compare the RS-based debris flow and field observation.

**Response #10:** Field observations were conducted by our co-author Dr. Noah Finnegan. However, these field excursions were not formal analyses, but rather were meant to confirm that the remotely sensed debris flow events indeed occurred. We are aware of an exhaustive field-based analysis in

the Dolan burn scar (Cavagnaro et al., 2021) and when that effort is complete, a comparison with the RS-based observations would be ideal.

**Line 262**. Simplifying the model description? WRF-Hydro has been used extensively.

**Response #11:** We agree that WRF-Hydro has been used extensively in simulating streamflow, but to our knowledge it has only rarely been employed in the landslide debris flow community. As it is a new tool in the debris flow, postfire hydrology, and natural hazards communities, we prefer to maintain our model description for the benefit of these newer audiences.

**Line 453,455**. Change unit of Ks to mm/hr.
**Response #12:** We prefer to keep the unit of Ks to be m/s because m/s is the unit that the Noah-MP LSM and WRF-Hydro utilize (Chen & Dudhia, 2001; see Table B3).

**Line 505-506.** Great metrics you used here in the context of hazards and their impacts.
**Response #13:** We thank the referee for their kind words.

**Line 669-672.** I agree the uniform precip introduce bias of hazards distribution over watersheds. But I highly suspect the assessments based on the data and method in this study, such as the computation grid size, MRMS quality in the study area, parameters on burned scars, and metrics (water volume rather than peak flow) used for debris flow assessment.

**Response #14:** It is unclear to us what the referee is asking here. We thank the referee for acknowledging their agreement that the use of uniform precipitation is one of the limitations of the USGS methods.

**Line 730-736.** You got the idea but confused why the volume of water is used for debris flow assessment.
**Response #15:** We thank the referee for identifying this confusing sentence. In this manuscript, we are not attempting simulate debris flow triggering processes. This sentence was intended to explain a future direction that could potentially advance our susceptibility assessment to better characterize debris flow occurrence likelihoods. To clear the confusion, we revised the sentence and the paragraph in our revised manuscript.

Lines 751 – 762 now read: A second capability in need of development is the use of WRF-Hydro to identify debris flow triggering time and location by employing a domain-specific rainfall ID threshold trained with historic landslide inventory and triggering rainfall events (Tognacca et al., 2000; Gregoretti & Dalla Fontana, 2007, 2008) or a newly developed dimensionless discharge and Shields stress threshold (Tang et al., 2019a; McGuire & Youberg, 2020). While in this study we do not attempt to simulate debris flow dynamics such as triggering, we note that WRF-Hydro is capable of simulating overland flow and streamflow at higher spatiotemporal resolutions [on scales that are similar to other debris flow mechanistic studies such as Rengers et al. (2016), McGuire et al. (2016, 2017), and Tang et al. (2019a, 2019b)]. Therefore, WRF-Hydro's capability to simulate the triggering processes of runoff-generated debris flows is potentially only limited by the spatiotemporal resolution of precipitation forcing and computing resources.

**Figure 4:** change rainrate unit to mm/hr, which is more common for post-fire hydrologic study. You may tune font size in all figures.

**Response #16:** We have changed the rainrate unit to mm/hr and tuned the font size for all figures in our revised manuscript.

**References:**

[revised manuscript text omitted]